# Where Do We Stand with Implicit Neural Representations? A Technical and Performance Survey

**Amer Essakine**[1,2*], **Yanqi Cheng**[1*], **Chun-Wun Cheng**[1], **Lipei Zhang**[1], **Zhongying Deng**[1], **Lei Zhu**[3], **Carola-Bibiane Schönlieb**[1], **Angelica I Aviles-Rivero**[4,1]

[1]*Department of Applied Mathematics and Theoretical Physics, University of Cambridge*
[2]*ENS Paris-Saclay*
[3]*ROAS & DSA, The Hong Kong University of Science and Technology (Guangzhou)*
[4]*Yau Mathematical Sciences Center, Tsinghua University*

**Reviewed on OpenReview:** https://openreview.net/forum?id=QTsJXSvAI2

## Abstract

Implicit Neural Representations (INRs) have emerged as a paradigm in knowledge representation, offering exceptional flexibility and performance across a diverse range of applications. INRs leverage multilayer perceptrons (MLPs) to model data as continuous implicit functions, providing critical advantages such as resolution independence, memory efficiency, and generalisation beyond discretised data structures. Their ability to solve complex inverse problems makes them particularly effective for tasks including audio reconstruction, image representation, 3D object reconstruction, and high-dimensional data synthesis. This survey provides a comprehensive review of state-of-the-art INR methods, introducing a clear taxonomy that categorises them into four key areas: activation functions, position encoding, combined strategies, and network structure optimisation. We rigorously analyse their critical properties—such as full differentiability, smoothness, compactness, and adaptability to varying resolutions—while also examining their strengths and limitations in addressing locality biases and capturing fine details. Our experimental comparison offers new insights into the trade-offs between different approaches, showcasing the capabilities and challenges of the latest INR techniques across various tasks. In addition to identifying areas where current methods excel, we highlight key limitations and potential avenues for improvement, such as developing more expressive activation functions, enhancing positional encoding mechanisms, and improving scalability for complex, high-dimensional data. This survey serves as a roadmap for researchers, offering practical guidance for future exploration in the field of INRs. We aim to foster new methodologies by outlining promising research directions for INRs and applications.

## 1 Introduction

Knowledge representation (Brachman, 2004) is a foundation in computational fields, playing a critical role in enabling systems to efficiently model, interpret, and manipulate information across various domains. Deep neural networks have demonstrated a powerful capacity for learning robust knowledge representation from data, and have become the predominant tools for addressing complex tasks in areas like computer vision (LeCun et al., 2015). The significance of effective knowledge representation extends beyond traditional methods, as it directly influences the performance and scalability of systems when handling diverse types of information such as images, video, and audio, whether in 1D, 2D, or 3D formats. Conventional approaches to encoding input signals typically rely on explicit discretisation, where the input space is segmented into distinct

---

*Equal contribution.

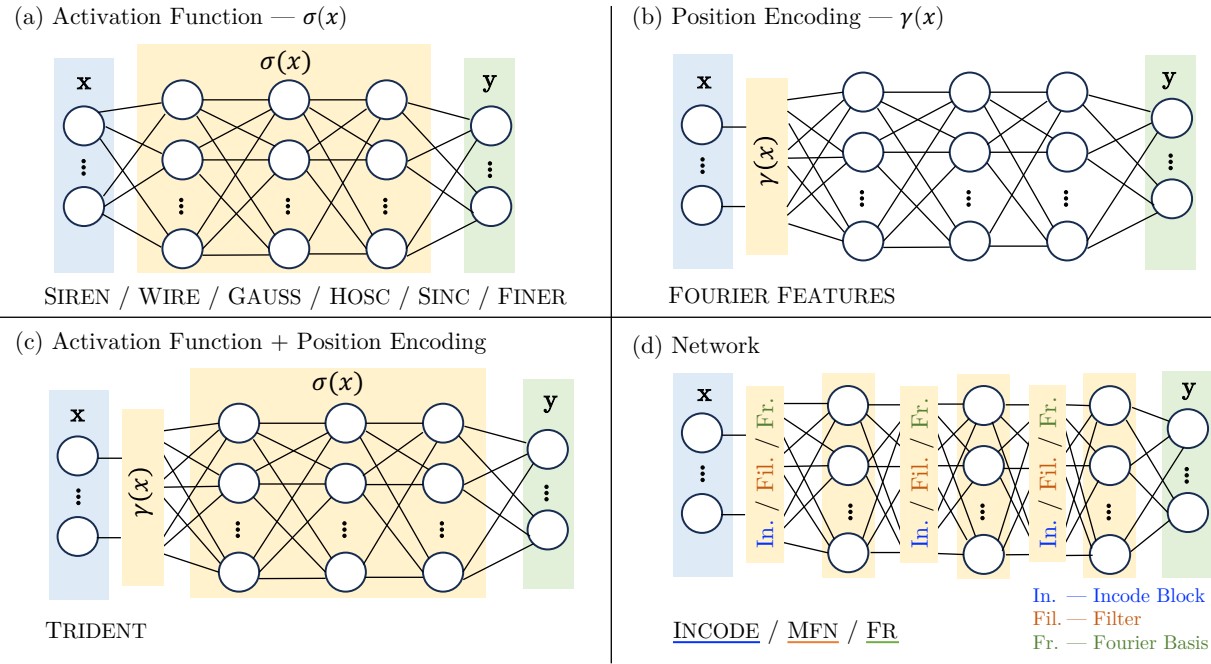

Figure 1: The four categories of state-of-the-art (SOTA) implicit neural representation (INR) methods. The yellow blocks highlight the specific components each method enhances. Specifically: (a) focuses on improving activation functions, (b) enhances position encoding, (c) integrates both (a) and (b) to simultaneously improve activation functions and position encoding, and (d) advances the overall network structure.

elements, such as point clouds (Achlioptas et al., 2018; Fan et al., 2017), voxel grids (Gadelha et al., 2017; Liao et al., 2018; Stutz & Geiger, 2018; Jimenez Rezende et al., 2016), and meshes (Kanazawa et al., 2018; Ranjan et al., 2018; Wang et al., 2018). While these methods can achieve adequate results, they present significant challenges when dealing with high-dimensional data (Mescheder et al., 2019). The computational cost of discretisation rises sharply with increasing dimensionality, making it inefficient, particularly for complex or irregular spaces. Moreover, traditional discretisation methods tend to require substantial memory, posing limitations for large-scale applications. Recent works have shown efficient representation of complex data, for instance latent representation, as seen in autoencoders and variational autoencoders (VAEs) (Tschannen et al., 2018), which encode signals into low-dimensional latent spaces through the use of neural networks, and structured dictionary learning (Chen et al., 2018; Tasissa et al., 2023), which seeks to approximate signals as sparse linear combinations of learned basis elements. Implicit Neural Representations (INRs) offer a promising alternative by using continuous functions to represent data, addressing many of the limitations of explicit methods, such as memory inefficiency and the high computational cost associated with discretisation.

The rise and continued development of INR have recently emerged as a new way to learn representation efficiently. Implicit representations differ from explicit (or discrete) representations by encoding information as a continuous generator function, which maps input coordinates to corresponding values within the defined input space, rather than directly storing feature or signal values. Consequently, there has been significant interest in utilising these networks as implicit functions, with notable success (Mildenhall et al., 2021; Mescheder et al., 2019; Xie et al., 2022). Specifically, Multi-Layer Perceptrons (MLPs) are trained to parameterise signals by taking coordinates as inputs, applying a mapping technique that projects this input into a higher-dimensional space, and predicting the associated data values at each coordinate. In this framework, the MLP functions as an Implicit Neural Representation, encoding the signal's information within its weights. For example, when applied to image data, pixel coordinates are fed into the MLP, which generates the corresponding RGB values, effectively learning a continuous, high-resolution representation of the image. However, the classic use of the ReLU activation function often resulted in suboptimal performance across many applications. To address this issue, reparametrised learning techniques (Rahaman et al., 2019)

have been first used to adjust the weights and mitigate bias, further enhancing the network's performance. (Fathony et al., 2020) presented a new architecture where the output of each layer is multiplied by a Gabor wavelet. Further researchers have introduced various activation functions. These include periodic sinusoidal functions (Sitzmann et al., 2020), time-frequency localised Gabor wavelets (Saragadam et al., 2023), Gaussian functions (Ramasinghe & Lucey, 2022), and the FINER network (Liu et al., 2024). Additionally, TRIDENT (Shen et al., 2023) is a network that integrates both positional encoding and a carefully chosen activation function.

Implicit neural functions have been further adjusted in various tasks, including image generation (Reddy et al., 2021), super-resolution (Wu et al., 2021; Chen et al., 2021), 3D object reconstruction (Chabra et al., 2020; Mescheder et al., 2019; Mildenhall et al., 2021), and modelling of complex signals (Xu et al., 2022). Its application on modeling 3D scenes has been developed as Neural Radiance Field scene representations (NeRFs) (Mildenhall et al., 2021) allowing for photorealistic image generation from multiple viewpoints. The use of multi-layer perceptrons (MLPs) for image and shape parameterisation provides distinct advantages. First, MLPs are resolution-independent as they operate within a continuous domain, enabling them to generate values for coordinates beyond pixel- or voxel-based grids. Thus, it improves performance in vision tasks. Second, their memory requirements are not constrained by signal resolution, allowing more memory-efficient representations compared to traditional grid or voxel methods (Huang et al., 2021; Park et al., 2019). The memory demand scales according to the complexity of the signal rather than the resolution. Additionally, MLPs address the limitations of locality biases often found in convolutional neural networks (CNNs), which can hinder generalisation (Chen & Zhang, 2019). Finally, MLP-based models are fully differentiable, offering adaptability across various applications (Zhang et al., 2024; Liu et al., 2020). Their weights can be optimised using gradient-based techniques, providing the flexibility needed for diverse tasks (Zhang et al., 2023; Xie et al., 2022; Tancik et al., 2020; Cheng et al., 2023). Another body of work is that of (Shenouda et al., 2024). The authors presented insights that may complement the broader discussions in this survey, particularly in exploring the potential of ReLU-based architectures for INR tasks.

Implicit Neural Representations (INRs) are increasingly intersecting with broader trends in Artificial Intelligence (AI). For instance, the adoption of INR in transformer architectures enables more expressive and scalable mechanisms for capturing complex data relationships (Chen & Wang, 2022). Another promising direction involves diffusion models that leveraging INR as guide (Hui et al., 2024) expanding their potential applications. Within this expanding landscape, INRs hold a unique position as compact, continuous representations of data, complementing other AI paradigms and offering a flexible tool for solving diverse problems in machine learning and beyond.

Despite the significant advancements in Implicit Neural Representations (INRs), there remains a notable gap in the literature that this work aims to address. The objective of this survey is to provide a comprehensive examination of diverse INR methods, offering both an in-depth analysis of their foundational principles and a thorough exploration of their wide-ranging applications. While progress has been made, existing reviews, such as the one by (Molaei et al., 2023) that focuses on medical applications, do not include experimental comparisons across different approaches, nor do they cover the full spectrum of use cases. Moreover, these works primarily address task-specific applications, rather than delving into the underlying technical principles that define INRs. As many of the tasks are built on the same core INR techniques, there is a need for a broader comparison that highlights differences in performance across various methodologies. This survey aims to bridge that gap by delivering an extensive review of INR methodologies and applications, while also offering a performance comparison to reveal their strengths and limitations. By doing so, we provide a more complete understanding of the field and its potential for future advancements.

To the best of our knowledge, this is the first survey paper to comprehensively explore both the fundamental and advanced functions of INRs through practically experimental comparisons across various applications, where we considered the SOTA methods listed in Figure 2. Our work serves as a systematic guide and roadmap for researchers, offering new perspectives on the capabilities of INR models. Additionally, we aim to inspire the broader research community to further investigate the potential of INRs across various domains. We believe this paper will help future exploration and innovation, encouraging deeper engagement with INR methodologies.

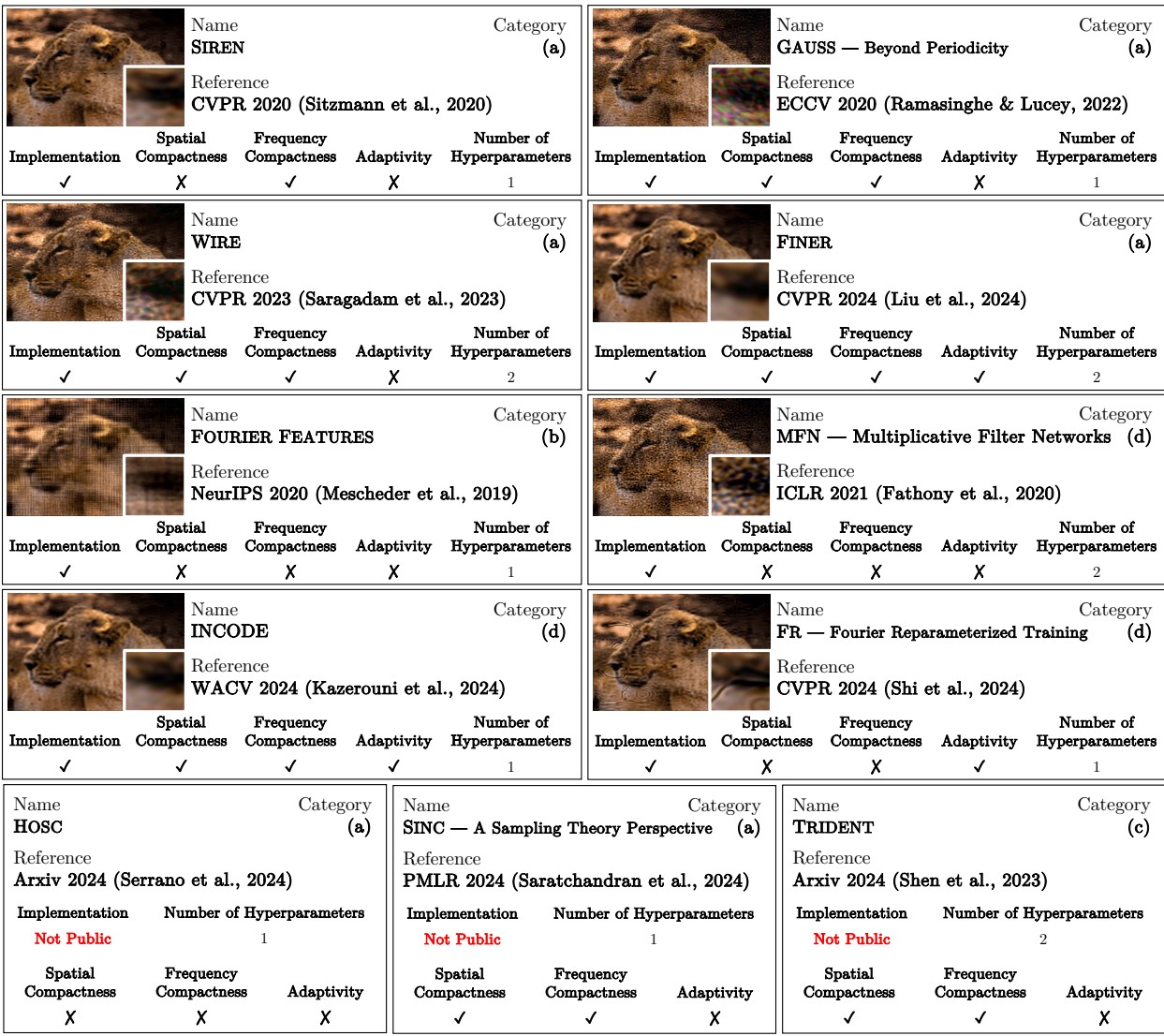

Figure 2: A comprehensive comparison of the INR methods, each represented by a method card. The cards outline key properties, including frequency compactness, spatial compactness, adaptability of the methods, and implementation details with the number of hyperparameters. The categories mentioned correspond to those in Figure 1.

**Contributions.** *This survey introduces a clear taxonomy of existing state-of-the-art (SOTA) INR techniques,* organising them into four key categories that represent critical advancements in the field (see Figure 1). First, methods in **the activation function** category (a) enhance the expressiveness and adaptability of INRs by improving activation functions, resulting in more flexible and capable representations. Notable examples include SIREN (Sitzmann et al., 2020), WIRE (Saragadam et al., 2023), GAUSS (Ramasinghe & Lucey, 2022), HOSC (Serrano et al., 2024), SINC (Saratchandran et al., 2024), and FINER (Liu et al., 2024), each offering distinct benefits in signal modelling through specialised activations. Second, **the position encoding** category (b), represented by techniques such as FOURIER FEATURES (Mescheder et al., 2019), focuses on refining how positional information is encoded into the model, enhancing the ability to capture fine-grained details in complex signals. Third, methods that combine **activation functions and position encoding** (c), like TRIDENT (Shen et al., 2023), address both aspects simultaneously, providing a more robust and flexible approach to representation learning. Finally, the network structure category (d), featuring techniques such as INCODE (Kazerouni et al., 2024), MFN (Fathony et al., 2020), and FR (Shi et al.,

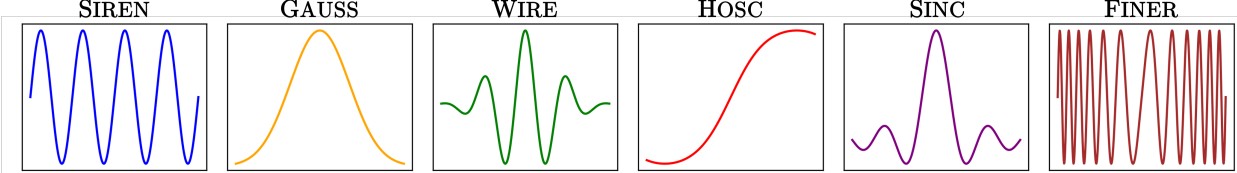

Figure 3: Visualisation of the activation functions used in the method, categorised as (a) in Figure 1.

2024), focuses on optimising the overall network architecture, incorporating additional components like in-coding blocks and filters to enhance learning and generalisation. These four categories collectively define the landscape of current INR research. Our work not only introduces this taxonomy but also further explores these foundational approaches through experimental analysis. Moreover, our survey offers valuable insights into why current methods are effective and highlights the key factors influencing performance trade-offs. In contrast to existing studies, we provide a comprehensive performance comparison across various inverse problem tasks.

## 2 Where Do We Stand with Implicit Neural Representations?

● **What Are Implicit Neural Representations?** Implicit neural representations consider the problem of finding a continuous representation of data. Given an input $x$, we are interested in learning a function that maps $x$ to a quantity of interest $a(x)$, while verifying an implicit equation depending on $a(x) : \mathbb{R}^{n_{in}} \to \mathbb{R}^{n_{out}}$ and possibly its derivatives:

$$F(x, a(x), \nabla_x a(x), \nabla_x^2 a(x), ...) = 0. \tag{1}$$

Our function of interest could be simply be the image, mapping the coordinates to their pixel values. To do that, we parametrise $a(x)$ as an MLP network with N layers, where the output of each layer is given by:

$$y_i = \sigma(W_i y_{i-1} + b_i), \tag{2}$$

where $\sigma$ is the activation function, $W_i \in \mathbb{R}^{n_i \hat{n}_{i-1}}$ are the weights, and $b_i \in \mathbb{R}^{n_i}$ the biases of the i-th layer. $x_0$ is the input coordinate (i.e., pixel coordinate for an image), and the final output reads: $y_M = W_N y_{N-1} + b_N$.

ReLU has been extensively utilised for function approximations in MLPs (Hanin & Rolnick, 2019). While they have demonstrated excellent performance across various applications, they often struggle to faithfully represent complex signals and capture fine details. This limitation arises because these networks tend to prioritise learning low-frequency components. Consequently, numerous approaches have been proposed in the literature to mitigate this bias.

● **General Properties of INRs.** Implicit Neural Representations (INRs) have several defining mathematical properties that distinguish them from traditional data representations. One key advantage is their continuous nature, where data is encoded not as discrete samples but as continuous functions. This allows INRs to represent high-resolution and fine-grained details without the need for large, memory-intensive storage, making them particularly efficient for high-dimensional data. The ability to interpolate between points seamlessly, without being constrained by a predefined resolution, is a core feature of this framework. The following list outlines some of the mathematical general properties that underpin INRs.

Continuity: INRs model data as continuous functions. For an input $x \in \mathbb{R}^{n_{in}}$, the output $a(x) \in \mathbb{R}^{n_{out}}$ is given by a continuous function $f : \mathbb{R}^{n_{in}} \to \mathbb{R}^{n_{out}}$, where: $f(x)$ is continuous over the domain of $x$.

Differentiability: INRs are fully differentiable, allowing optimisation via gradient descent. Given the function $f(x)$ representing the data, the derivatives of $f$ with respect to the input $x$ exist and are continuous: $\nabla f(x)$ and $\nabla^2 f(x)$ exist and are continuous.

Resolution Independence: The output of INRs is not tied to the resolution of the data. For any input coordinate $x$, INRs can generate corresponding values without a predefined grid structure, i.e., they operate on a continuous domain: $f(x)$ can produce outputs for any $x \in \mathbb{R}^{n_{\text{in}}}$, independent of any fixed resolution.

Compactness: The representation of data through INRs is compact in memory usage, as the function $f(x)$ is encoded within the weights of a neural network, rather than as a large array of discrete samples. If the model uses $N$ parameters to define $f(x)$, memory usage scales with $N$ instead of the data resolution: Memory usage scales as $O(N)$, where $N$ is the number of parameters in the network.

Frequency Adaptivity: Through techniques such as positional encoding or adaptive activation functions, INRs can capture a wide range of frequencies in the data. For input $x$, INR models can handle both low-frequency and high-frequency components: $f(x)$ can represent both low-frequency and high-frequency signals, depending on model design.

Smoothness of Representations: INRs tend to produce smooth representations by virtue of their construction. The activation functions, such as sinusoidal or wavelet-based functions, ensure that the output is smooth and continuous across the input domain: $f(x)$ is typically smooth over its input domain.

Scalability: INRs scale well to complex, high-dimensional tasks. The memory and computational requirements grow with the complexity of the function $f(x)$, not with the dimensionality of the output space: Computational complexity scales with the complexity of $f(x)$, not directly with the data dimensionality.

## 2.1 Taxonomy INR: (a) Activation Function

To begin, we present our first category in our taxonomy and review the existing literature. One of the critical challenges in Implicit Neural Representations (INRs) is overcoming the bias towards learning low-frequency components, commonly observed in neural networks that use traditional activation functions such as ReLU (Hanin & Rolnick, 2019). The ReLU function, while effective in many settings, has a discontinuous nature that limits its ability to capture higher-frequency signals, making it suboptimal for tasks that require detailed or high-resolution data representation. However, recent work Shenouda et al. (2024) has demonstrated that by imposing specific constraints inspired by B-spline wavelets, ReLU-based networks can overcome this limitation, enabling them to approximate high-frequency components effectively while maintaining their computational efficiency. Furthermore, alternative activation functions such as sigmoid or tanh, while continuous, are not expressive enough to adequately capture the full range of frequency components necessary for complex tasks.

To address these limitations, several novel activation functions have been proposed in the literature, specifically designed to overcome the low-frequency bias. These functions are engineered to better capture a wider range of frequencies, enabling more detailed and accurate representations in INR models. We next explore some of the key activation functions that have been introduced, highlighting their contributions to advancing the performance of INRs.

Activation Function — $\sigma(x)$

SIREN / WIRE / GAUSS / HOSC / SINC / FINER

● **SIREN (Sitzmann et al., 2020)** The authors in (Sitzmann et al., 2020) proposed SIREN using the sine function as an activation function:

$$\sigma(x) = \sin(\omega x), \tag{3}$$

where $\omega$ is a scaling hyperparameter chosen tailored for each task to modulate the low-frequency bias. This activation function stands out due to its periodic nature, enabling it to model a broad spectrum of frequencies effectively. The periodic properties of SIREN allow the network to represent high-frequency variations with ease, making it particularly suitable for tasks requiring intricate details, such as image and audio reconstruction, and solving complex differential equations. One key benefit of the sine-based activation function is its smoothness, which not only facilitates stable gradient flow during training but also allows for

explicit calculation of derivatives. This capability makes SIREN highly effective for applications involving inverse problems, where precise derivative computation is critical. Additionally, the smooth, continuous representation that SIREN offers enables it to generalise well across varying resolutions and achieve high-quality, resolution-independent outputs. The authors proposed a specialised weight initialisation scheme to maintain the distribution of activations throughout the network layers, which ensures efficient convergence and avoids issues like vanishing or exploding gradients. This initialisation technique further enhances SIREN's capacity to capture fine-grained details while remaining memory efficient.

However, despite these advantages, the SIREN activation function also has certain limitations. Due to its reliance on a fixed periodic function, it can struggle to faithfully represent complex high-frequency details, which may result in artifacts or reduced accuracy in applications with highly detailed or non-smooth features. Furthermore, SIREN's reliance on a single frequency scaling parameter can limit its flexibility in adapting to diverse signal characteristics, making it less versatile for some complex tasks.

● **GAUSS FUNCTION** (**Ramasinghe & Lucey, 2022**) When exploring beyond periodicity, the network employs the Gaussian function as the activation function, given by:

$$\sigma(x) = \exp(-(sx)^2), \tag{4}$$

where $s$ controls the width of the frequency. The Gaussian activation function offers several benefits, particularly its smoothness, which makes it well-suited for applications requiring continuous and localised representations. Because it is localised in both the spatial and frequency domains, it is advantageous in tasks where smooth transitions and localised details are important, such as in denoising or the reconstruction of smooth objects. However, a notable drawback is its lack of periodicity, which limits its effectiveness in representing high-frequency signals. This makes the Gaussian activation function less suitable for tasks that involve intricate or oscillatory data, such as audio reconstruction or high-detail image synthesis, where capturing fine details at higher frequencies is crucial. The balance between its smoothness and the inability to handle high-frequency information defines its strengths and weaknesses, depending on the application at hand.

● **WIRE** (**Saragadam et al., 2023**) The authors proposed using the Gabor wavelet as an activation function, defined as:

$$\sigma(x) = \exp(-(sx)^2 + i\omega x), \tag{5}$$

where the imaginary part is represented by the $i$-term. The Gabor wavelet is effective in minimising the product of its standard deviations in both the time and frequency domains, allowing it to represent features across both space and frequency. This dual localisation enables the function to capture fine-grained details and frequency components more effectively. As a result, the Gabor wavelet combines the advantages of both the SIREN network, which excels in high-frequency representations, and the Gaussian network, known for its smooth, spatial localisation.

While the Gabor wavelet offers strong representational capabilities, a potential drawback is its relatively higher computational cost compared to simpler activation functions. The need to process both spatial and frequency components simultaneously can lead to increased complexity in network training and inference, making it less efficient in scenarios where computational resources or speed are critical. In addition, the Gabor wavelet's ability to capture fine-grained details across both space and frequency, while advantageous, may also increase the risk of overfitting, particularly when dealing with noisy or sparse data. The model may become overly sensitive to minor variations in the input, making it less generalisable to unseen data. While most of the INR activation functions only require one hyperparameter, WIRE requires two hyperparameters ($s$ and $\omega$), which in turn results in more computational effort to choose the appropriate hyperparameters.

● **HOSC** (**Serrano et al., 2024**) Similar to the SIREN network, the authors in (Serrano et al., 2024) proposed the HOSC function, defined as:

$$\sigma(x) = \tanh(\beta \sin(x)), \tag{6}$$

where $\beta$ is a sharpness factor that controls the steepness and behavior of the hyperbolic function.

The HOSC function inherits several properties from SIREN, particularly in terms of computational efficiency and low memory usage. Additionally, higher values of $\beta$ enable the capture of fine details, especially due to

abrupt amplitude changes at $x = n\pi$, while lower values of $\beta$ emphasize lower-frequency components, similar to SIREN's behavior. The authors further proposed a network architecture where $\beta$ increases across layers, enabling the model to effectively represent both high and low-frequency signals. They also introduced Ada-HOSC, an alternative version where $\beta$ is treated as a learnable parameter, which further leverages the simple derivative form of the HOSC function for improved adaptability. Whilst HOSC function offers flexibility, it has several potential disadvantages. The sharpness factor $\beta$ must be carefully tuned to balance the capture of high- and low-frequency components, and improper tuning could lead to unstable training or convergence issues. Additionally, as $\beta$ increases across layers, the computational complexity and training time may also increase, potentially slowing down the overall process.

● **SINC FUNCTION** (Saratchandran et al., 2024) Motivated by the classic Shannon sampling theorem and drawing similarities between INR and sampling, the authors in (Saratchandran et al., 2024) proposed the sinus cardinal as an activation function, which form is given by:

$$\sigma(x) = \text{sinc}(\omega x). \tag{7}$$

The SINC is localised in the space domain, but not in the frequency domain. It also forms a generating system, meaning the set of its translates $\{x \to \text{sinc}(x - k) | k \in \mathbb{Z}\}$ can be used for sampling, enabling the network to approximate signals and represent high-order features effectively. The SINC activation function, motivated by sampling theory, offers several distinct advantages. One of its primary strengths is its localisation in the space domain, which makes it particularly effective at representing spatial information. Additionally, the SINC function can generate a set of translates that approximate signals, making it well-suited for tasks that involve high-order feature representation. Its connection to sampling theory provides a theoretical foundation for reconstructing signals, which enables accurate approximation of continuous data.

However, a key limitation of the SINC is its lack of localisation in the frequency domain. This means it might struggle to represent or capture high-frequency variations as effectively as other functions designed with periodicity or high-frequency adaptivity. As a result, while the SINC excels in certain applications where spatial representation is paramount, it may not be the best choice for tasks that require precise handling of complex, high-frequency features.

This trade-off between space localisation and frequency adaptability highlights its strengths in certain contexts, while also pointing to its limitations in handling tasks involving detailed frequency components.

● **FINER** (Liu et al., 2024) The SIREN model struggles to represent a broad spectrum of frequencies due to its reliance on a fixed scaling value. This limitation restricts the range of reconstructed frequencies, which may be insufficient, particularly when the frequency distribution of the signal is unknown or varies significantly. To overcome this issue, the authors of the work (Liu et al., 2024) proposed an activation function given by:

$$\sigma(x) = \sin(\omega(|x| + 1)x). \tag{8}$$

This activation function is both smooth and periodic, inheriting all the properties of the sine function. In addition, the scaling parameter $\omega$ varies dynamically across the network based on the bias of different nodes. Higher bias values correspond to rapidly varying functions that can capture high frequencies, while lower bias values correspond to slower variations, effectively representing lower frequencies. This dynamic scaling provides the FINER approach with greater flexibility to represent a broader range of frequencies. Specifically, nodes with higher bias can capture high-frequency details, while nodes with lower bias are better suited for representing lower-frequency information. This adaptability offers FINER a major advantage over fixed-scaling models like SIREN, as it is able to represent a much broader range of frequencies, particularly when the frequency distribution is variable or unknown. Accordingly, the authors proposed to initialise the bias coefficients as an uniform distribution:

$$b \sim U(-k, k). \tag{9}$$

Choosing a sufficiently large value of $k$ ensures that the set of frequencies captured by the model is not constrained by the initialisation.

However, there are also potential disadvantages to consider. The reliance on dynamic bias scaling, while powerful, can introduce complexity in tuning and initialisation. The effectiveness of FINER depends on

choosing an appropriate range for the bias coefficients, which may require careful experimentation. Moreover, if the initial bias range is not selected carefully, the model might not be able to capture the full spectrum of frequencies required for certain tasks, potentially limiting its generalisation abilities in some cases.

## 2.2  Taxonomy INR: (b) Positional Encoding

**FOURIER FEATURE (Tancik et al., 2020)** Another approach to learn high-frequency features involves mapping the input coordinates to a higher-dimensional space using a Fourier feature mapping $\gamma$. This enables the network to effectively represent these features, addressing limitations in capturing fine details. Several encoders have been proposed in the literature (Tancik et al., 2020), including:

- **Basic**: $\gamma(x) = [\cos(2\pi x), \sin(2\pi x)]^T$. This simple encoding projects the input into two oscillatory components, providing a basic yet effective way to introduce periodicity and high-frequency information into the network.

- **Positional Encodings**: $\gamma(x) = [x, \dots, \cos(2\pi\omega^{j/m}x), \sin(2\pi\omega^{j/m}x)]^T$, where $\omega$ represents the frequency hyperparameter and $m$ the embedding size. This type of encoding enhances the network's capacity to capture both fine and coarse features, making it highly effective for tasks involving spatial data including computer vision applications.

- **Random Fourier Features**: $\gamma(x) = [\cos(2\pi Bx), \sin(2\pi Bx)]^T$, with $B$ being a random Gaussian matrix sampled from $\mathcal{N}(0, \omega^2)$. In this approach, the transformation matrix $B$ is sampled from a Gaussian distribution $\mathcal{N}(0, \omega^2)$. Random Fourier features offer a stochastic way to approximate complex functions, and they are particularly useful in applications that benefit from randomness, such as kernel approximation or uncertainty modelling.

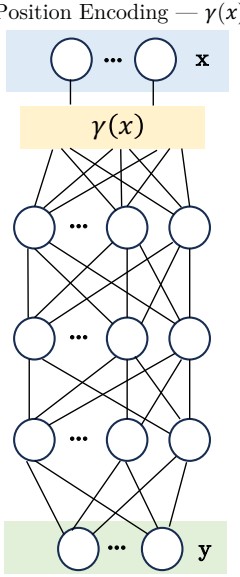

Position Encoding — $\gamma(x)$

**FOURIER FEATURES**

After applying the Fourier feature mapping to the input, the resultant features are fed into a Multi-Layer Perceptron (MLP) using ReLU activation functions.

## 2.3  Taxonomy INR: (c) Activation Function + Position Encoding

**TRIDENT (Shen et al., 2023)** Trident is a mix between the two approaches of

Activation Function $\sigma(x)$
+
Position Encoding $\gamma(x)$

using an activation function, and applying a Fourier mapping. The first layer of the network is expressed as:

$$y_1 = W_1\gamma(x) + b_1, \qquad (10)$$

where $\gamma$ is the Fourier features mapping, which is formulated as $\gamma(x) = [x, \dots, \cos(2\pi\sigma^{j/m}x), \sin(2\pi\sigma^{j/m}x)]^T$.

The subsequent layers output are computed as:

$$y_i = \sigma(W_i y_{i-1} + b_i), \qquad (11)$$

where $\sigma(x) = exp(-sx^2)$ is the activation function. This activation function serves three essential purposes. Firstly, the network can effectively represent high-order features. This property is ensured by expressing the exponential function as an infinite series of cosine functions:

$$\exp(-\cos(x)^2) = \sum_{n=1}^{\inf} A_n \cos(2x)^n. \qquad (12)$$

TRIDENT

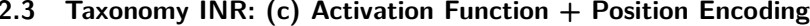

Secondly, the activation function also enables Trident to represent a broad spectrum of frequencies. This capability is demonstrated by expressing the cosine function as a series of shifted and scaled components:

$$\cos(x)^n = \frac{1}{2^n}\binom{n}{\frac{n}{2}} + \frac{2}{2^n}\sum_{k=0}^{n}\frac{n}{2}\binom{n}{k}\cos((n-2k)x). \tag{13}$$

This formulation ensures that the activation function can span a wide frequency range, enhancing the model's capacity to learn complex patterns. Finally, TRIDENT ensures compact spatial localisation by selecting the coefficients $A_n$ according to a Gaussian window. This configuration helps the network focus on specific regions in the input space, improving its ability to capture fine-grained spatial features.

### 2.4 Taxonomy INR: (d) Network

● INCODE (Kazerouni et al., 2024) Another approach that enhances the SIREN architecture involves using an activation function of the form:

$$\sigma(x) = a\sin(b\omega x + c) + d, \tag{14}$$

where $a$ controls the amplitude, determining the overall strength of the activation function; $b$ manages frequency scaling, influencing the range of frequencies the network can capture; $c$ introduces a phase shift, shifting the waveform horizontally; and $d$ sets the vertical shift, adjusting the baseline or "brightness" of the signal. In this formulation, $\omega$ serves as a fixed hyperparameter that defines the base frequency of the sinusoid, while $b$ acts as a learnable parameter, allowing the network to tune the frequency dynamically.

A key highlight of the INCODE approach is that these parameters—$a$, $b$, $c$, and $d$—are not fixed but are predicted dynamically by a separate module known as the harmoniser network. This enables the model to adjust the characteristics of the activation function in real time, resulting in more adaptive and expressive representations. With this flexibility, the network is better equipped to capture a wider range of patterns, both high-frequency and low-frequency, making it suitable for tasks that involve complex and multi-scale data.

The intuition behind this design lies in the ability to modulate the signal dynamically, mimicking the behavior of real-world signals that exhibit varying frequencies and amplitudes. By learning to adjust these parameters during training, the network can efficiently model data with varying characteristics, such as audio signals with fluctuating pitch or images with diverse levels of detail. This approach offers several advantages. The dynamic modulation of frequency and amplitude provides the network with enhanced expressiveness, allowing it to generalise better across tasks that involve varying input patterns. Moreover, the flexibility in phase and baseline shifts ensures that the

Network

In. / Fil. / Fr.

In. / Fil. / Fr.

In. / Fil. / Fr.

In. — Incode Block
Fil. — Filter
Fr. — Fourier Basis

INCODE / MFN / FR

network can align and normalize data effectively, further improving its ability to learn complex relationships. However, this increased flexibility also introduces additional complexity. The harmoniser network adds computational overhead, and the need to carefully tune multiple parameters may make training more challenging. Additionally, while the dynamic nature of the activation function enhances expressiveness, it may increase the risk of overfitting, especially when applied to small or noisy datasets.

● MFN–Multiplicative Filter Networks (Fathony et al., 2020) The Multiplicative Filter Networks (MFN) architecture offers an alternative to traditional Multi-Layer Perceptrons (MLPs) by replacing them with a sequence of Hadamard products and nonlinearities. In this approach, the output at each layer is computed by applying a nonlinear activation function to the input, followed by a Hadamard (element-wise) product between the transformed input and the weight matrix. This design introduces a new way to model complex interactions within the data, leveraging multiplicative relationships rather than the additive ones typically found in MLP architectures.

| INR | Baseline Complexity | Extra Complexity |
|-----|---------------------|------------------|
| SIREN | | The sinusoidal operation is computed in constant time, with no additional complexity beyond $O(1)$. |
| GAUSS | | The Gaussian function involves squaring the input, multiplying, and applying the exponential function, all of which are $O(1)$. |
| WIRE | $O(1)$ | Combines Gaussian $O(1)$ operations with cosine $O(1)$, resulting in no additional complexity beyond $O(1)$. |
| HOSC | | Combines sine $O(1)$ with hyperbolic tangent $O(1)$. The sharpness factor $\beta$ introduces no additional complexity. |
| SINC | | Involves sine and division operations. The division introduces negligible overhead, remaining $O(1)$. |
| FINER | | The sinusoidal operation with dynamic scaling based on $|x|$ introduces no additional asymptotic complexity, as absolute value and addition are $O(1)$. |

Table 1: Computational Complexity of Activation Functions in INR Methods.

The MFN framework supports two specific types of activation functions. The first is the sinus nonlinearity, defined as:

$$\sigma(x; \theta^i) = \sin(\omega^i x + \phi^i), \tag{15}$$

where in the $i$th layer, the parameter $\theta^i$ includes the frequency $\omega^i$ and phase shift $\phi^i$ allow the model to capture oscillatory patterns effectively. The second is the Gabor wavelet, given by:

$$\sigma(x; \theta^i) = \exp(-\gamma^i \|x - \mu^i\|_2^2) \sin(\omega^i x + \phi^i), \tag{16}$$

which introduces both spatial and frequency localisation with an additional parameter, the scale term $\gamma^i$. The combination of these functions enables MFN to model a wide range of data patterns by approximating signals as a mixture of sinusoidal and wavelet components. One of the key advantages of MFN is their ability to represent complex, high-order details that are often challenging for standard architectures like SIREN or Fourier Feature Networks (FOURIER FEATURES). By using multiplicative interactions and flexible nonlinearities, MFN can capture intricate dependencies within the input data. This makes them particularly effective in applications requiring fine-grained representation and high-frequency details, such as audio synthesis, image reconstruction, and other tasks involving implicit neural representations.

However, the MFN architecture also comes with certain challenges. The use of Hadamard products and complex activations can increase the computational cost compared to simpler activation functions like SIREN. Training MFN can require more careful hyperparameter tuning, as the interplay between frequency, phase, and spatial parameters can be sensitive to initialisation. Moreover, while the multiplicative design enhances the model's expressiveness, it may also increase the risk of overfitting.

● **FR–Fourier Reparameterised Training (Shi et al., 2024)** Another approach to bypass the low-frequency bias is to use an appropriate weight reparametrisation. The idea is to reparametrise each row of the weight matrix of the $i$-th layer $W_i$ as a sum of Fourier bases. The output of each layer is given by:

$$y_i = \sigma(\Lambda_i B_i y_{i-1} + b_i) \tag{17}$$

Where $\Lambda_i$ is a learnable weight matrix and $B_i \in \mathbb{R}^{M \cdot n_{i-1}}$ is the Fourier basis matrix given by

$$B_i^{k,l} = \cos(w_k z_l + \phi_k) \tag{18}$$

for $i = 1, ..., M$ and $j = 1, ..., n_{out}$. We denote $B_i^{k,l}$ as the entries of $B_i$, $(z_l)$ which is the sampling position sequence, and $M$ as the number of Fourier basis considered.

The authors proposed to choose $P$ phases as an array of phase shifts, which are evenly distributed over the interval from $[0, 2\pi]$, and to take $2F$ frequencies, consisting of a low-frequency basis $\{\frac{1}{F}, \frac{2}{F}, ..., 1\}$ and a high frequency basis $\{1, 2, ..., F\}$, which gives a total of $M = 2FP$ bases. $F$ and $P$ are hyper parameters and should be chosen for each task independently.

As for the sampling sequence, the number of sampling points is the number of features, while the points were

chosen to be sampled uniformly from the interval $[-\pi F, \pi F]$. Finally, it was proposed to draw the weights of the learnable matrix $\Lambda_i$ following the distribution:

$$\Lambda_i^{k,l} \sim \mathcal{U}\left(-\sqrt{\frac{6}{\alpha}}, \sqrt{\frac{6}{\alpha}}\right), \tag{19}$$

where $\alpha = M \sum_{t=1}^{d_{n-1}} (B_i^{l,t})^2$.

While Fourier Reparameterised Training (FR) offers a powerful approach to mitigating low-frequency bias, it also presents certain challenges and disadvantages. A key limitation lies in the increased complexity of the model. Reparameterising the weight matrices using Fourier bases introduces additional hyperparameters, such as the number of frequency components $F$, the number of phase shifts $P$, and the sampling sequences. These hyperparameters must be carefully selected for each specific task, which can require extensive experimentation and tuning, increasing the burden on practitioners.

Another disadvantage involves the computational overhead. Since each layer's weight matrix is reparameterised as a sum of Fourier bases, the model becomes more computationally intensive compared to simpler architectures. This increased complexity can lead to longer training times and higher memory usage, which may limit the applicability of FR models in resource-constrained environments or real-time applications.

### 2.5 On the Complexity of INRs

The Table 1 provides an overview of the computational complexity associated with various activation functions used in implicit neural representations (INRs). All the functions listed share a baseline computational complexity of $O(1)$, which means that evaluating these functions for a single input requires constant time. This characteristic is crucial for INR methods, as they often deal with high-dimensional data where computational efficiency is paramount. Each activation function in the table demonstrates $O(1)$ complexity, but they vary in the operations they involve and their ability to handle specific tasks. For instance, the SIREN function relies solely on a sine operation, making it straightforward and efficient with no additional computational overhead. This simplicity makes it well-suited for representing periodic and oscillatory patterns in data. It can be proven from the results that it is the second fastest, 482 seconds, among all tested methods for the dosing task reported in Table 3. The Gaussian (GAUSS) activation function, while also $O(1)$, requires squaring the input, multiplying it by a scalar, and applying an exponential operation. These operations remain computationally lightweight, with empirically reported 709 seconds for denoising task in Table 3, yet they provide the function with the ability to represent smooth and spatially localised features, which is especially useful for tasks involving smooth transitions. WIRE takes this further by combining the Gaussian operation with a cosine function. Although this combination adds slight computational overhead compared to standalone functions like SIREN or GAUSS (as verified in Table 3 for the denoising task, 1485 seconds, which is double the computational time of GAUSS and triple that of SIREN), it still retains the $O(1)$ complexity. The dual spatial and frequency localisation capabilities provided by WIRE justify this marginal increase in computational effort, making it versatile for representing more complex patterns. HOSC builds on a sine function by applying a hyperbolic tangent transformation. While this adds an extra step, both the sine and hyperbolic tangent functions are computed in constant time, maintaining $O(1)$ complexity. The flexibility introduced by the sharpness factor $\beta$ allows HOSC to adapt between high- and low-frequency signal representations without compromising computational efficiency. The SINC, inspired by the sinus cardinal, involves both a sine computation and a division operation. Although the division introduces a minor computational overhead, it is negligible and does not affect the overall $O(1)$ complexity. SINC's strong spatial localisation capabilities make it particularly valuable for signal reconstruction tasks, though its frequency adaptability may be more limited than other functions.

## 3 Experimental Results

In this section, we present a comprehensive comparison of the state-of-the-art implicit neural representation (INR) methods that have publicly available implementations. These methods were evaluated across a diverse set of tasks, ranging from 1D audio reconstruction to various image reconstruction tasks, including denoising,

super-resolution, and CT reconstruction, as well as a 3D column occupancy task. This broad evaluation aims to highlight the strengths and limitations of each method in different application domains, providing a holistic view of their capabilities and performance across multiple dimensions.

For a fair comparison of all implemented INR methods, we implemented each approach using the default hyperparameters, per tasks, recommended in the respective papers. All models were trained and tested on an RTX 4070 GPU with 8 GB of RAM.

### 3.1 1D Applications: Audio Reconstruction

The task of audio reconstruction involves approximating the function $f : \mathbb{R} \to \mathbb{R}$, which represents the audio signal. We aim to represent 6 seconds of a musical piece by Bach. For all approaches, we use a 5-layer neural network and apply a scaling factor of 30 to the first layer to better capture the high-frequency components typically present in audio signals. The experiments are run for 2000 iterations.

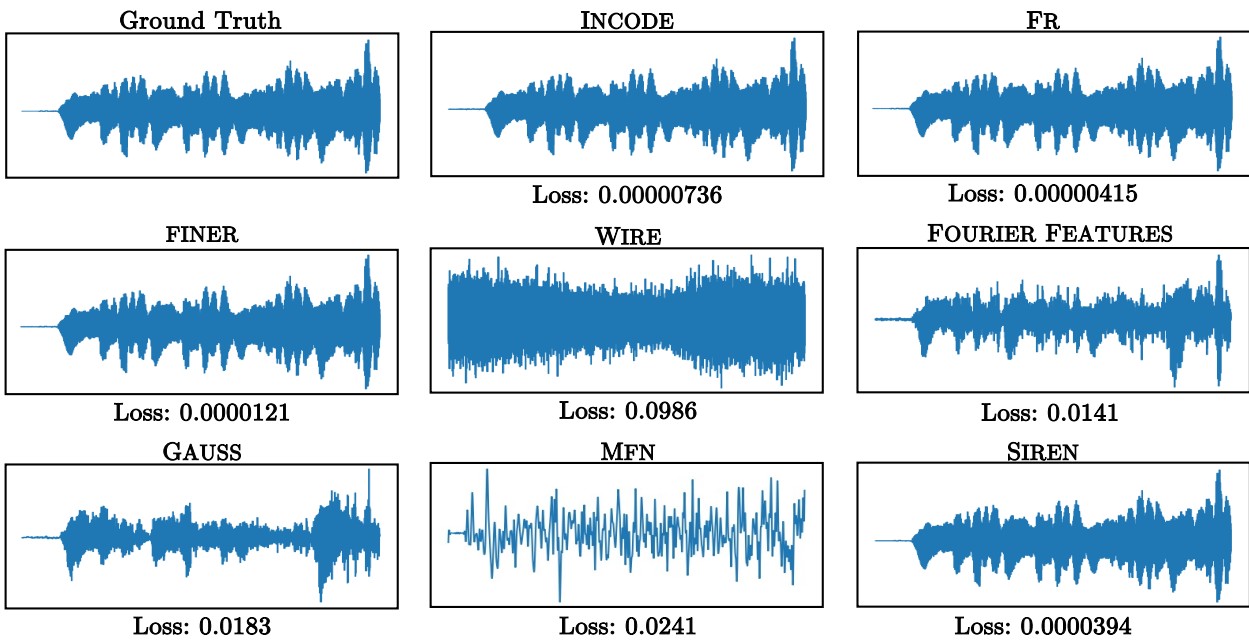

Figure 4: The visualisation with $L_2$ loss metric comparison on audio reconstruction task across the 8 implicit neural representation methods.

Figure 4 indicates the overall performance among all the methods of audio reconstruction. This comparison highlights the performance of various implicit neural representation (INR) methods in reconstructing an audio signal, with FR (Fourier Reparameterised Training) and INCODE showing the best results. FR achieves the lowest loss due to its effective use of Fourier reparameterisation, which captures both low- and high-frequency components, overcoming the low-frequency bias typical of many models. INCODE also performs exceptionally well, thanks to its dynamic scaling of frequencies, which allows it to adapt to the complexities of the signal. SIREN and FINER follow closely, leveraging periodic activation functions and frequency scaling to handle oscillatory patterns effectively.

In contrast, methods like WIRE and MFN struggle to capture the intricacies of the waveform, resulting in noisier outputs. WIRE's Gabor wavelet, while useful for localised frequency representations, fails to generalise across the entire frequency spectrum of the audio, while MFN's multiplicative structure is more challenging to tune for this task. FOURIER FEATURES and GAUSS offer moderate performance, with the former limited by fixed feature sets and the latter by the absence of periodicity, making them less capable of capturing high-frequency details.

Table 2: Comparison of the implicit neural representation methods on PSNR (Peak Signal-to-Noise Ratio) and SSIM (Structural Similarity Index Measure) for CT image reconstruction across varying numbers of projections (20, 50, 100, 150, 200, and 300).

| Method | 20 projections | | 50 projections | | 100 projections | |
|---|---|---|---|---|---|---|
| | PSNR↑ | SSIM↑ | PSNR↑ | SSIM↑ | PSNR↑ | SSIM↑ |
| INCODE (Kazerouni et al., 2024) | 24.38 | 0.651 | 27.33 | 0.762 | 31.48 | 0.890 |
| FR (Shi et al., 2024) | 25.58 | 0.738 | 29.14 | 0.852 | 31.18 | 0.917 |
| FINER (Liu et al., 2024) | 25.53 | 0.731 | 28.20 | 0.841 | 30.92 | 0.888 |
| WIRE (Saragadam et al., 2023) | 20.73 | 0.417 | 25.01 | 0.651 | 28.83 | 0.826 |
| FOURIER FEATURES (Tancik et al., 2020) | 25.64 | 0.780 | 26.44 | 0.803 | 26.74 | 0.802 |
| GAUSS (Ramasinghe & Lucey, 2022) | 22.21 | 0.548 | 26.44 | 0.752 | 27.80 | 0.764 |
| MFN (Fathony et al., 2020) | 21.10 | 0.402 | 22.70 | 0.487 | 25.30 | 0.643 |
| SIREN (Sitzmann et al., 2020) | 20.85 | 0.421 | 24.42 | 0.607 | 29.61 | 0.842 |

| Method | 150 projections | | 200 projections | | 300 projections | |
|---|---|---|---|---|---|---|
| | PSNR↑ | SSIM↑ | PSNR↑ | SSIM↑ | PSNR↑ | SSIM↑ |
| INCODE (Kazerouni et al., 2024) | 33.94 | 0.939 | 34.29 | 0.946 | 34.76 | 0.953 |
| FR (Shi et al., 2024) | 30.25 | 0.907 | 30.49 | 0.910 | 30.40 | 0.910 |
| FINER (Liu et al., 2024) | 31.84 | 0.914 | 32.13 | 0.922 | 32.24 | 0.927 |
| WIRE (Saragadam et al., 2023) | 30.54 | 0.891 | 31.88 | 0.916 | 30.95 | 0.902 |
| FOURIER FEATURES (Tancik et al., 2020) | 26.98 | 0.804 | 26.94 | 0.803 | 26.94 | 0.794 |
| GAUSS (Ramasinghe & Lucey, 2022) | 27.85 | 0.766 | 27.89 | 0.773 | 27.90 | 0.848 |
| MFN (Fathony et al., 2020) | 28.29 | 0.793 | 30.50 | 0.868 | 33.63 | 0.935 |
| SIREN (Sitzmann et al., 2020) | 31.58 | 0.907 | 32.02 | 0.918 | 32.71 | 0.928 |

## 3.2 2D Inverse Problems: Image Reconstruction

We conducted a series of experiments on various inverse problems in 2D, including CT reconstruction, image denoising, and single-image super-resolution, to evaluate the performance and robustness of the proposed methods.

### 3.2.1 CT Reconstruction

CT reconstruction is a process used in medical imaging to create detailed cross-sectional images of the body from multiple X-ray projections taken at different angles. Implicit neural representations (INR) can reconstruct an image using a given number of measurements by minimising the following loss function:

$$\mathcal{L} = \|\text{sinogram}(output) - \text{sinogram}(truth)\|, \tag{20}$$

where the sinogram represents the transformation of the image into a set of projections. In all methods considered, we focus on the problem of recovering an image from varying numbers of measurements. Specifically, we conduct experiments with 20, 50, 100, 150, 200 and 300 measurements. We employ a two-layer neural network with 300 hidden features and train it for 5000 iterations.

Both Figure 5 and Table 2 indicate that the reconstruction quality improves for all methods as the number of projections increases, which aligns with the general expectation that more projections provide richer information, leading to more accurate and detailed reconstructions. Observing from Table 2, INCODE demonstrates superior performance, achieving the highest PSNR (Peak Signal-to-Noise Ratio) and SSIM (Structural Similarity Index Measure) values in all scenarios, particularly excelling at higher projection counts with scores of 34.76 dB for PSNR and 0.953 for SSIM at 300 projections. At lower projection counts, FR stands out, delivering the best results with a PSNR of 29.14 and SSIM of 0.852 at 50 projections, and further improving to 31.18 PSNR and 0.917 SSIM at 100 projections, making it highly effective with limited data. FINER performs competitively as the number of projections increases, while WIRE and MFN consistently lag behind, exhibiting the lowest scores across all scenarios. Moderate performers like FOURIER FEATURES and GAUSS

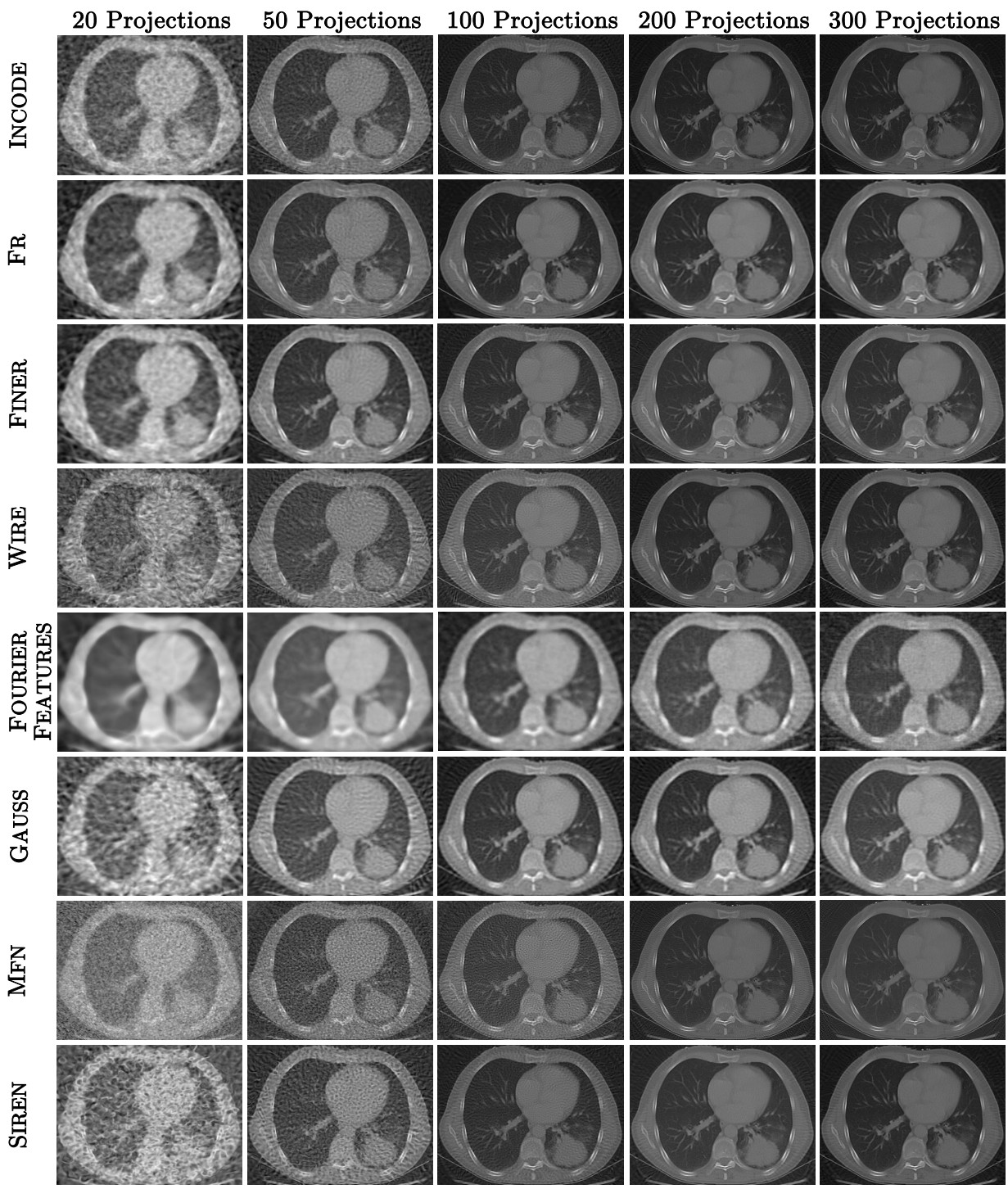

Figure 5: The visual comparison of CT reconstruction results using the 8 implicit neural representation methods across 20, 50, 100, 200, and 300 projections.

show adequate results in low to medium projection settings but are outperformed by INCODE and FR at higher counts. These findings underscore the strengths and limitations of each method, highlighting INCODE as the most robust approach for high-quality CT reconstruction, especially when more projection data is available.

In Figure 5, INCODE and FR consistently produce the clearest reconstructions across all projection settings, with INCODE showing superior noise suppression and finer structural details, even at lower projections such as 20 and 50. FINER, while performing well at higher projection counts, exhibits noticeable artifacts and lower fidelity at lower projection numbers, particularly at 20 projections. WIRE and GAUSS struggle significantly in low-projection scenarios, with severe blurring and structural distortions that persist even at 100 projections, highlighting their limited ability to capture fine details. FOURIER FEATURES achieves moderate performance, performing better than WIRE and GAUSS but failing to reach the level of clarity achieved by INCODE and FR. At high projections (200 and above), all methods show considerable improvement, but differences remain evident; for instance, MFN and SIREN display sharper boundaries and fewer artifacts at 300 projections, yet they still lag behind INCODE in overall image quality. This visual comparison underscores the robustness of INCODE and FR across different projection settings, while revealing the limitations of other methods, particularly in scenarios with sparse data.

When considering the properties across all methods, INCODE stands out due to its balanced combination of spatial and frequency compactness, which is reflected in its superior performance across all CT reconstruction tasks, particularly at higher projection counts. Methods such as FR and FOURIER FEATURES excel in CT reconstruction due to their strength in frequency-based representations, allowing them to effectively capture fine details and structural information, especially when the number of projections is moderate to high. On the other hand, methods like SIREN and GAUSS, which lack adaptivity and frequency compactness, struggle to achieve comparable performance, particularly at lower projection counts where these properties are crucial for maintaining image quality. WIRE and FINER, despite having adaptivity and more flexibility due to their higher number of hyperparameters, show limitations in handling high-frequency details, leading to suboptimal reconstructions in sparse projection settings. Thus, the results indicate that while adaptivity and spatial compactness are beneficial, the inclusion of frequency compactness and a balanced set of hyperparameters are critical factors for achieving high-quality CT reconstructions, as exemplified by the robust performance of INCODE, FR, and FOURIER FEATURES across different projection scenarios.

### 3.2.2 Image Denoising

Image denoising involves recovering the original, noise-free image from a noisy observation. We evaluate the different approaches by introducing two types of noise: Poisson noise, which typically arises in low-light conditions and photon-limited imaging, and Gaussian noise, which is common in electronic sensor noise and general environmental interference. We employ a two-layer neural network with 256 hidden features and train it for 2000 iterations.

We report the results in Figure 6 and Table 3.

The results of the denoising task highlight the strong performance of INCODE, FR, and FINER, which achieve the best balance between denoising quality and fine detail preservation. INCODE has the highest PSNR (29.63) and provides a visually sharp reconstruction, closely matching the ground truth. However, it comes at the cost of significant computational time (2370 seconds), making it less practical for real-time applications. In contrast, FR and FINER offer similarly high-quality results with PSNR values of 29.47 and 29.05, respectively, but with much lower computation times (752 and 643 seconds), making them more efficient for practical use.

SIREN also performs well, with a reasonable trade-off between accuracy (PSNR 28.60) and computational speed (482 seconds). On the other hand, FOURIER FEATURES, while the fastest (403 seconds), sacrifices reconstruction quality, evident in its lower PSNR (27.16) and noisier output. Methods like WIRE, GAUSS, and MFN fall behind in both visual quality and PSNR, with their outputs appearing either over-smoothed or retaining significant noise, making them less competitive for high-quality denoising tasks.

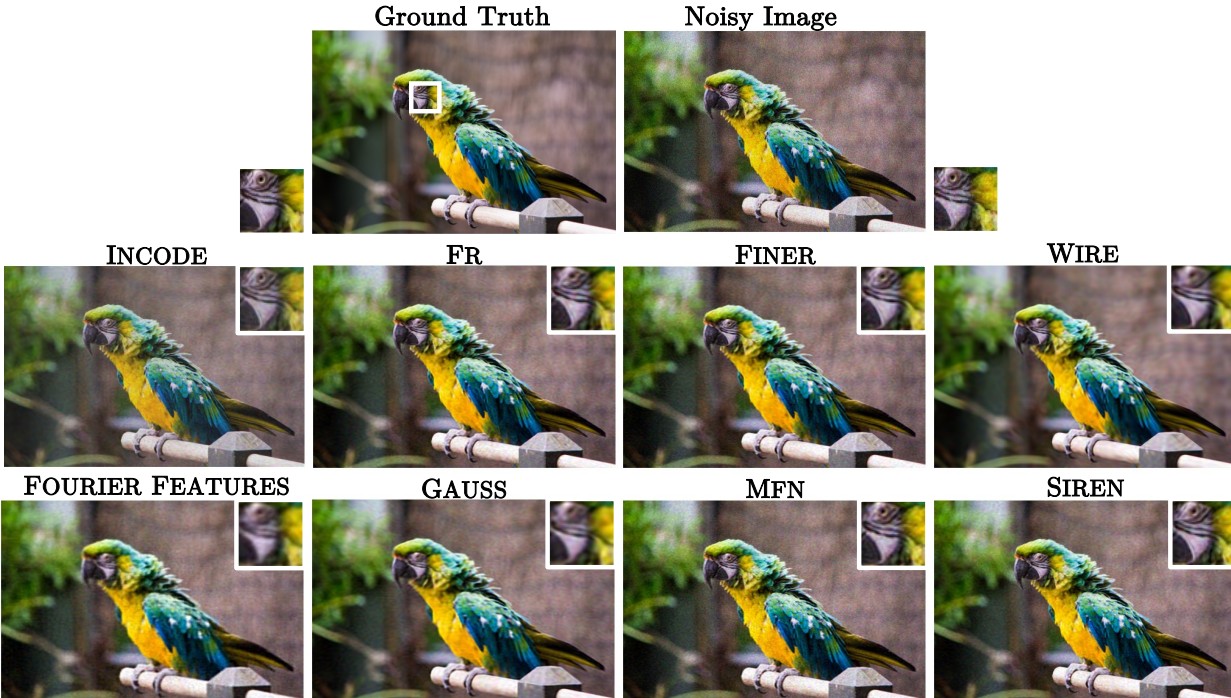

Figure 6: The visual comparison across the 8 implicit neural representation methods on the denoising task.

Table 3: The metric comparison of the PSNR performance, and the time that the 8 implicit neural representation methods used for the same number of iterations.

| METHOD | INCODE | FR | FINER | WIRE | FOURIER FEATURES | GAUSS | MFN | SIREN |
|---|---|---|---|---|---|---|---|---|
| **PSNR ↑** | 29.63 | 29.47 | 29.05 | 28.74 | 27.16 | 28.09 | 28.22 | 28.60 |
| **Time(s) ↓** | 2370 | 752 | 643 | 1485 | 403 | 709 | 1280 | 482 |

That is, FR and FINER emerge as the most balanced methods, providing high-quality reconstructions with manageable computation times, while INCODE is ideal when quality is prioritised over speed. FOURIER FEATURES, though fast, is less suitable for tasks requiring fine detail preservation.

### 3.2.3 Single Image Super Resolution

In the super resolution task, we aimed to enhance the quality of images by reconstructing high-resolution visuals from low-resolution inputs. We conducted experiments at four distinct resolution scales: 2×, 4×, 8×, and 16×.

In Table 4, we observe that the performance of methods varies significantly across different upscaling factors. For the 2× and 4× resolutions, INCODE,and FR show superior performance in terms of all Peak Signal-to-Noise Ratio (PSNR), Structural Similarity Index (SSIM), and Learned Perceptual Image Patch Similarity (LPIPS) scores, indicating their capability to capture fine details and maintain structural integrity at these moderate upscaling factors. As the resolution factor increases to 8× and 16×, FR and FINER emerges as the top performer, showcasing its robustness to high upscaling, which is the more challenging cases. Conversely, GAUSS and MFN struggle at higher upscaling factors, as evidenced by their relatively low SSIM and LPIPS values, indicating that these methods are less effective at reconstructing high-frequency details. The performance gap becomes more significant as the resolution increases, highlighting their limited capacity for handling higher upscaling tasks. On the other hand, while SIREN and FOURIER FEATURES do not perform as well as the other methods across all four scales, they exhibit less variation in performance as the resolution

Table 4: The metric evaluations across PSNR, SSIM and LPIPS of image super resolution task with 2×, 4×, 8×, and 16× resolutions.

| Method | 2× | | | 4× | | |
|---|---|---|---|---|---|---|
| | PSNR↑ | SSIM↑ | LPIPS ↓ | PSNR↑ | SSIM↑ | LPIPS ↓ |
| INCODE (Kazerouni et al., 2024) | 29.56 | 0.896 | 0.176 | 27.43 | 0.816 | 0.422 |
| FR (Shi et al., 2024) | 29.10 | 0.879 | 0.243 | 27.49 | 0.822 | 0.371 |
| FINER (Liu et al., 2024) | 29.50 | 0.892 | 0.191 | 27.44 | 0.818 | 0.395 |
| WIRE (Saragadam et al., 2023) | 28.91 | 0.874 | 0.252 | 25.93 | 0.754 | 0.447 |
| FOURIER FEATURES (Tancik et al., 2020) | 26.31 | 0.767 | 0.428 | 25.73 | 0.733 | 0.473 |
| GAUSS (Ramasinghe & Lucey, 2022) | 28.08 | 0.851 | 0.324 | 24.10 | 0.681 | 0.619 |
| MFN (Fathony et al., 2020) | 29.28 | 0.890 | 0.203 | 24.99 | 0.716 | 0.610 |
| SIREN (Sitzmann et al., 2020) | 29.00 | 0.877 | 0.241 | 27.27 | 0.811 | 0.409 |

| Method | 8× | | | 16× | | |
|---|---|---|---|---|---|---|
| | PSNR↑ | SSIM↑ | LPIPS ↓ | PSNR↑ | SSIM↑ | LPIPS ↓ |
| INCODE (Kazerouni et al., 2024) | 25.43 | 0.731 | 0.597 | 22.91 | 0.638 | 0.715 |
| FR (Shi et al., 2024) | 23.75 | 0.662 | 0.643 | 23.40 | 0.682 | 0.637 |
| FINER (Liu et al., 2024) | 25.69 | 0.743 | 0.544 | 23.39 | 0.667 | 0.665 |
| WIRE (Saragadam et al., 2023) | 21.72 | 0.558 | 0.703 | 18.06 | 0.422 | 0.773 |
| FOURIER FEATURES (Tancik et al., 2020) | 22.31 | 0.549 | 0.473 | 20.65 | 0.518 | 0.702 |
| GAUSS (Ramasinghe & Lucey, 2022) | 20.06 | 0.464 | 0.872 | 17.14 | 0.349 | 0.868 |
| MFN (Fathony et al., 2020) | 19.49 | 0.411 | 0.750 | 16.61 | 0.290 | 0.934 |
| SIREN (Sitzmann et al., 2020) | 24.66 | 0.703 | 0.583 | 19.09 | 0.509 | 0.780 |

increases, suggesting a consistent, though lower, reconstruction capability that is less sensitive to the increase in upscaling factors.

While observing from the visualisation results, in Figure 7, 8, 9, and 10, the conclusion coincide with the table result with more detailed features. From the zoom in view of Figure 7, 8 for the lower resolution task, we observe INCODE and FINER successfully reconstructed the detailed fur at the animal's ear and the mustache. However, as the scaling increases, INCODE starts to show color artifacts in the reconstruction result, while FINER oversmooths the details. This oversmoothing is likely a result of FINER's lack of spatial compactness, which prevents it from preserving fine features at higher resolutions. In fact several other methods has shown artifacts with different features when the super resolution scale increases. For example, the Fourier based methods that lack of compact frequency and spatial representations shows undesirable repetitive patterns at finer scales, which FR exhibits twisting pattern blobs, and FOURIER FEATURES shows the grid like artifacts. Similarly, noise artifacts become more alleviated in the reconstruction results of WIRE, MFN, GAUSS, and SIREN all of which lack the task-specific parameters necessary for tuning to different levels of super-resolution scales. This adaptability proves to be beneficial, as both the visualisations and the metrics show that INCODE, FINER, and FR which has this task-specific parameters perform better across different scales.

### 3.3 3D applications: Occupancy Reconstruction

The 3D occupancy task involves representing a 3D shape by determining whether points in space are inside or outside the object. For the dragon mesh, a complex model with 2,748,318 points and 5,500,000 triangles, we aim to capture its detailed geometry by classifying points as either inside (1) or outside (0) the shape. The final reconstructed model has 566,098 vertices and 1,132,830 triangles, and the task evaluates how well the occupancy representation matches the true shape.

We can observe in Table 5 that FINER stands out in the Intersection over Union (IoU) metric, with a value of 0.00064 higher than the second-performing method, INCODE. Given the mesh size, this difference is significantly notable.

The visualisation comparison can be viewed in Figure 11, where all methods successfully reconstruct the global structure of the occupancy. However, the zoomed-in view indicates variations in texture among the

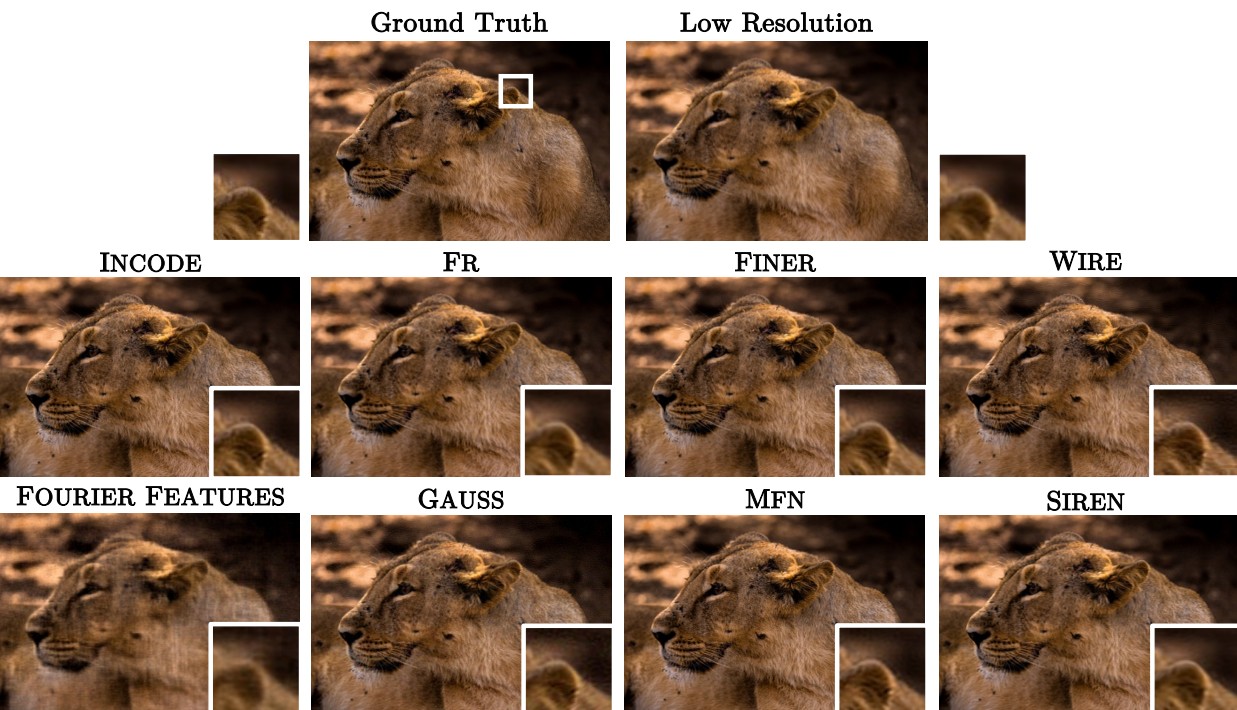

Figure 7: The visualisation comparison of the 8 INR methods on super resolution task with 2×.

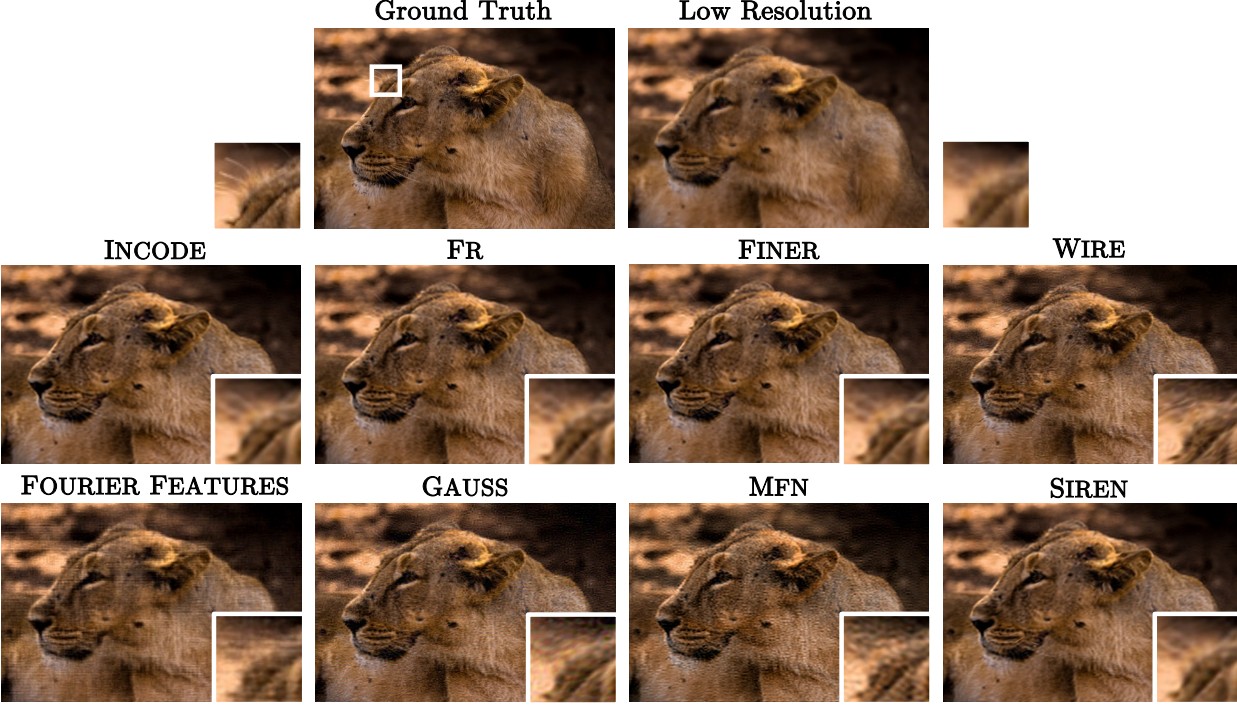

Figure 8: The visualisation comparison of the 8 INR methods on super resolution task with 4×.

reconstruction results. In particular, MFN reconstructed a non-smooth surface with significant deformation in the local structure. It is worth noting that MFN, along with FR and FOURIER FEATURES, which are the

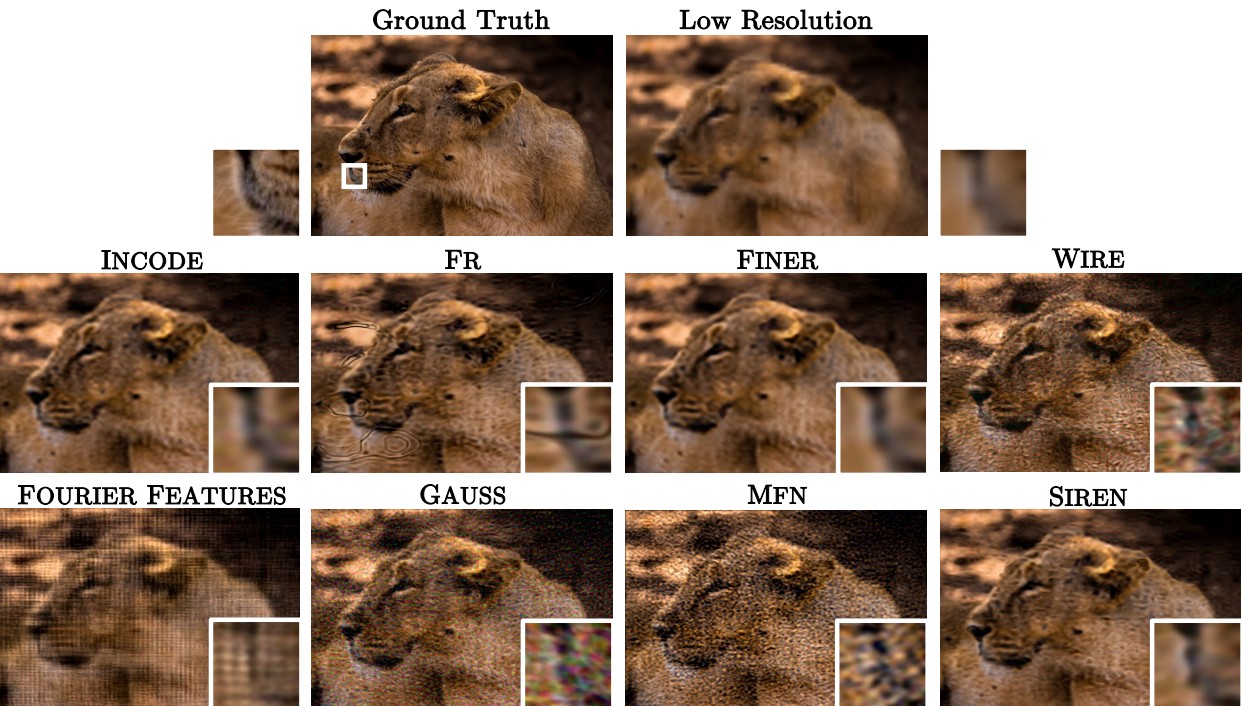

Figure 9: The visualisation comparison of the 8 INR methods on super resolution task with 8×.

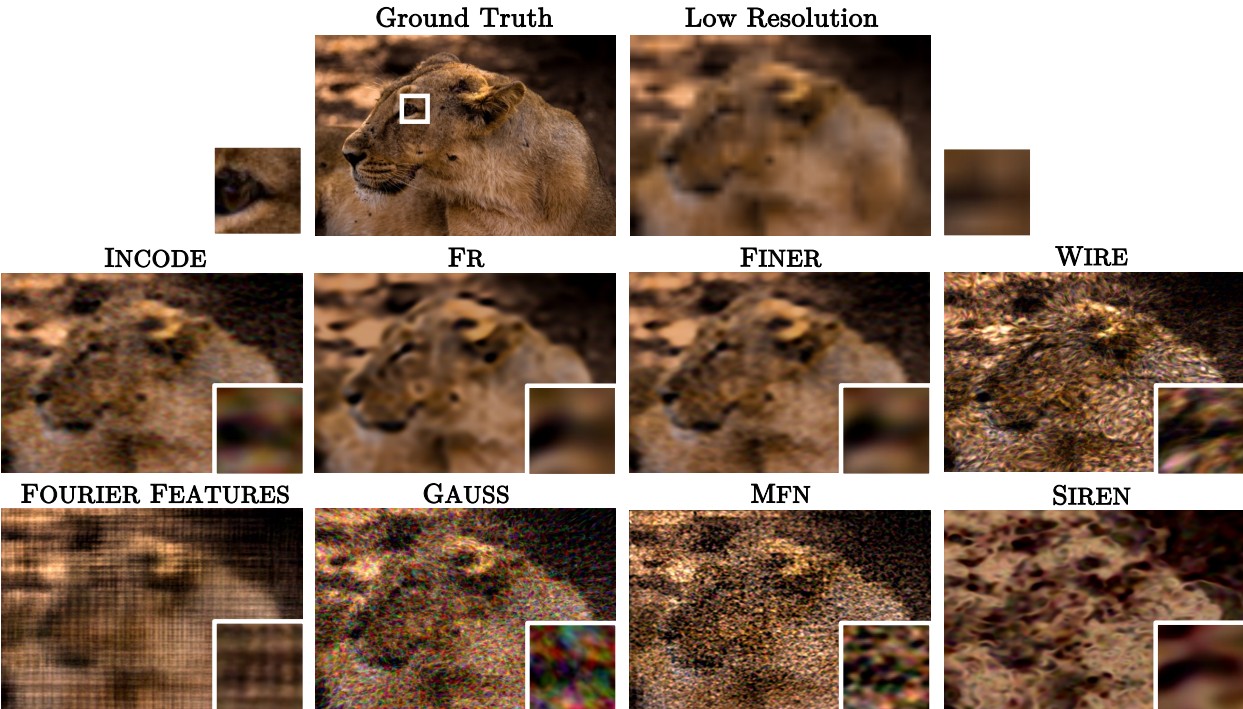

Figure 10: The visualisation comparison of the 8 INR methods on super resolution task with 16×.

only 3 methods lacking frequency compactness, performed the worst among all 8 compared methods. This

Table 5: The metric comparison of 3D occupancy task across the 8 INR methods.

| METHOD | INCODE | FR | FINER | WIRE | FOURIER FEATURES | GAUSS | MFN | SIREN |
|---|---|---|---|---|---|---|---|---|
| **IoU** ↑ | 0.99564 | 0.99136 | 0.99628 | 0.99454 | 0.99424 | 0.99510 | 0.97540 | 0.99552 |

highlights the importance of frequency compactness when reconstructing finely detailed surfaces in the 3D occupancy task.

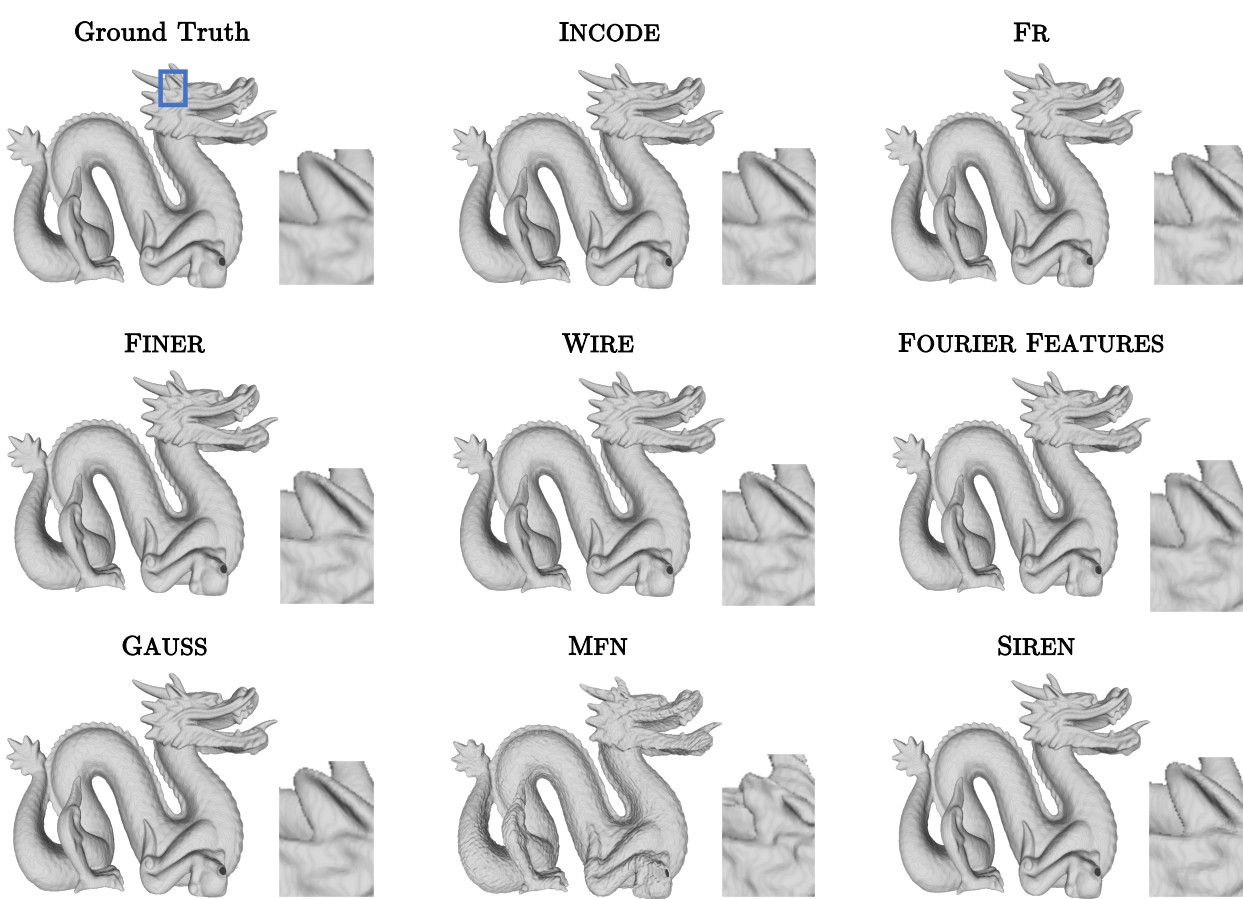

Figure 11: The visualisation comparison of the 8 INR methods on 3D shape representation task.

## 4 Discussion & Future Work

The results from the experiments highlight that different INR methods vary in performance based on the tasks they are applied to, such as image denoising, super-resolution, and audio reconstruction. INCODE consistently stands out as a top performer across several tasks, with high PSNR scores and superior ability to capture fine details, especially in challenging tasks like CT image reconstruction and super-resolution. However, INCODE comes with a drawback of significantly high computational cost, making it less feasible for real-time applications where efficiency is crucial. This is a clear trade-off between quality and time efficiency that practitioners need to consider.

FR (Fourier Reparameterised Training) also demonstrates excellent performance, particularly due to its ability to handle a broad spectrum of frequencies through Fourier reparameterisation, addressing the low-frequency bias inherent in many models. It strikes a good balance between accuracy and computational

efficiency, offering a viable alternative to INCODE for many tasks, especially when real-time performance is a concern. FINER is another high-performing model, with its dynamic scaling feature allowing it to adapt to varying frequency distributions. This makes it highly effective across multiple applications, though like FR, it requires careful tuning of hyperparameters. SIREN, while effective in capturing high-frequency oscillatory patterns, falls slightly behind in tasks that require broader frequency adaptability, though its relatively fast training time makes it suitable for practical implementations. Lower-performing methods like WIRE, GAUSS, and MFN struggle with both accuracy and generalisation. These methods show limitations in handling complex high-frequency information or fine detail preservation, particularly in tasks like super-resolution or denoising where these features are critical.

For applications where computational efficiency is less critical and the highest reconstruction quality is needed, INCODE would be the preferred choice. However, for real-time applications or cases where a balance between quality and speed is crucial, FR and FINER offer strong alternatives with their adaptable frequency handling and more manageable computation times. SIREN can be a good option when tasks require periodic signal reconstruction but with less variability in frequency content. One area where INR methods still face challenges is scalability, particularly in handling extremely high-resolution or highly detailed tasks. Future work could explore more efficient frequency encoding mechanisms that reduce the computational overhead without sacrificing quality. Additionally, optimising activation functions to be more adaptive to task-specific requirements could improve generalisation across diverse datasets and applications. Enhancing the dynamic adaptability of models like FINER could allow for more robust performance across tasks that involve varying levels of detail, potentially improving their usability in real-world scenarios.

Finally, focusing on methods that efficiently integrate both spatial and frequency compactness while reducing computational demands should be a priority for future research. Such advancements would make INR models more accessible for real-time applications while maintaining high fidelity in complex tasks like image and audio synthesis. While the surveyed methods provide a comprehensive and well-structured taxonomy of implicit neural representations (INRs), there are several areas present opportunities for deeper exploration to enhance our understanding of this domain. For instance, while the interplay between activation functions and position encoding is acknowledged, further investigation into how these components synergise could reveal additional avenues for optimisation. Similarly, potential approaches beyond the proposed taxonomy, such as hybrid models that integrate INRs with explicit representations, offer a promising direction for enhancing performance in tasks requiring both adaptability and precision, such as 3D reconstruction. Furthermore, the influence of hyperparameters, including encoding frequencies, network architectures, and activation scaling, remains a pivotal factor in fine-tuning method effectiveness across various data modalities. By building on the insights provided in this survey, future work can explore these areas to refine and expand the utility of INR techniques in a broader range of applications. The development of implicit neural representations (INRs) has the potential to transform various domains positively. However, it is crucial to consider ethical implications and ensure their responsible use to avoid unintended societal consequences.

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
