# OpenReview forum: "Where Do We Stand with Implicit Neural Representations? A Technical and Performance Survey"
_TMLR — Accepted by TMLR_

### Review · Reviewer_7Zht · 2024-11-28

**Summary Of Contributions:**

This work provides a survey of Implicit Neural Representations, including a taxonomy of INR technique improvements in literature as well as experimental comparisons.  The taxonomy identifies 4 categories of advancements: activation functions, position encoding, combined strategies, and network structure optimization. For each category, approaches from relevant works are summarized and compared on the basis of their mathematical properties, and then empirically evaluated across multiple applications for 1D, 2D, and 3D signals. The work concludes with a discussion including recommendations for which approach to use, given task characteristics and computational efficiency constraints.

**Audience:**

Yes

**Broader Impact Concerns:**

There is no included broader impact statement. One could be included (either separately or as a paragraph in the finally section) to strengthen the work, discussing topics on ethical issues surrounding INRs such as deepfakes.

**Claims And Evidence:**

Yes

**Requested Changes:**

I recommend this work for acceptance upon making the following changes.

As all of the activation functions presented in Section 2 are univariate, providing a plot for each (as a single figure) would help your audience visualize and compare them. This would be particularly useful for the less well-known functions like sinc as well as uncommon combinations of functions such as the HOSC activation function.

For each experiment and each method, you used the default hyperparameter settings presented in their respective paper. While I agree this is a simple fair setting, it would be worth explicitly adding this caveat to your discussion of the results of each experiment, such as MFN not capturing higher frequencies in section 3.1.

Although the final section is labeled “Discussion and Future Work”, only the final paragraph (just over 3 lines of text) discusses future work in INRs. What holes can you identify between the surveyed methods? Are there (potential) approaches outside of the taxonomy? How do hyperparameters (or lack thereof) affect the utility of a given approach, and how could this be amended in further development? What is the interplay between INRs and other trends like transformers, NeRFs, etc.? For example, can the weights of an INR network be generated directly by another model rather than trained from scratch? These later questions could also be supplemented with discussion in Section 1 to frame where INRs already fit within the context of AI.

Specific typo/error corrections for clarity:

Page 2: “by taking input coordinates by a mapping technique that projects the network input into a higher-dimensional space. It predicts the associated data values.” -> “by taking coordinates as inputs, applying a mapping technique that projects this input into a higher-dimensional space, and predicting the associated data values at each coordinate.”

Page 4: “...with N layer, …” -> “...with N layers, …”

Page 5: “The ReLU function […] has a discontinuous nature…” -> “The ReLU function […] has a non-smooth nature…”

Page 6: “The authors in (Sitzmann et al., 2020) proposed SIREN…” and “The authors propose using the Gabor wavelet”: please use consistent tense (either always past or always present). Searching through the document for “author” shows a lot of inconsistent tense usage.

Page 6: “When exploring beyond periodicity, the network employs…” sounds like the network is exploring beyond periodicity. You could rather say something like “In non-periodic activation functions, the authors propose using the Gaussian function, defined as: …”. Later in your evaluation of this technique, you say “it is advantageous in tasks where … localised details are important” but then say it is “less suitable for tasks that involve intricate [...] data”, but these seem contradictory.

Page 8: “...the authors to initialise…” -> “...the authors proposed to initialise…”

Page 10: Please define $i, \theta_i, \gamma_i,$ and $\mu_i$. Also, Equation 16 has an extraneous left parenthesis.

Page 10: “...appropriate weight reprametisation of weights.” -> “...appropriate weight reparameterisation."

Page 11: “...that has publicly…” -> “...that have publicly…”

**Strengths And Weaknesses:**

Overall, the paper is well-written, technically sound, and informative, providing a valuable resource for researchers interested in INRs, both new and experienced.

The experiments cover a diverse set of tasks, demonstrating the capabilities and challenges of different INR methods in practice. The results and discussion can both help practitioners select the appropriate approach for their task as well as help guide new research in the field.

The included literature is up-to-date, focusing on works from the last ~4 years.

Weakness and suggested changes to improve them are included in the next section.

---

> ### Author Response · Authors · 2024-12-27
> **Official Response 1**
>
> ➡️ **I recommend this work for acceptance upon making the following changes....As all of the activation functions presented in Section 2 are univariate, providing a plot for each (as a single figure) would help your audience visualize and compare them. This would be particularly useful for the less well-known functions like sinc as well as uncommon combinations of functions such as the HOSC activation function.**
>
> Thank you for your suggestion. We agree that providing visualisations for the activation functions discussed in Section 2 would greatly assist in comparing their behaviors and help clarify their differences, especially for the less well-known functions like sinc and the HOSC activation function.
>
> In response to your suggestion, we have generated a figure that includes plots of all the  activation functions mentioned. This includes standard functions like SIREN, GAUSS, WIRE, and FINER, as well as the sinc and HOSC functions. The plots will be presented together in a single figure, with clear labelling and axes for comparison.
>
> We have added the following figure (now Figure 3) to the revised manuscript, which compares the activation functions and highlights the distinct characteristics of the sinc and HOSC functions, as requested.
>
> ➡️ **For each experiment and each method, you used the default hyperparameter settings presented in their respective paper. While I agree this is a simple fair setting, it would be worth explicitly adding this caveat to your discussion of the results of each experiment, such as MFN not capturing higher frequencies in section 3.1.**
>
> Thank you for your valuable comment. We would like to clarify that we used the default hyperparameters for each method because the original authors have already identified and optimised these parameters to achieve the best possible performance for their respective methods. Our choice of these default settings was made to ensure consistency with the established settings used in the original papers.
>
> Regarding Section 3.1, our focus is not on the hyperparameter choices but on the inherent limitations of the MFN method itself. Specifically, we wanted to highlight that MFN struggles to capture higher frequencies in the task due to the constraints of its underlying conceptual framework, rather than due to suboptimal hyperparameter settings. We will make sure to explicitly mention this in the revised manuscript to avoid any potential confusion.
>
>
> ➡️ **Although the final section is labeled “Discussion and Future Work”, only the final paragraph (just over 3 lines of text) discusses future work in INRs. What holes can you identify between the surveyed methods? Are there (potential) approaches outside of the taxonomy? How do hyperparameters (or lack thereof) affect the utility of a given approach, and how could this be amended in further development? What is the interplay between INRs and other trends like transformers, NeRFs, etc.? For example, can the weights of an INR network be generated directly by another model rather than trained from scratch? These later questions could also be supplemented with discussion in Section 1 to frame where INRs already fit within the context of AI.**
>
> We thank the reviewer for this insightful suggestion. We agree that expanding the discussion on future work in INRs would enhance the manuscript. In the revised version, we include a more detailed discussion addressing the gaps between surveyed methods, potential approaches outside the proposed taxonomy, and the influence of hyperparameters on the utility of various methods.
>
> Additionally, we provide further discussion in blue in Section 1 in the revised version that explores the interplay between INRs and broader trends, such as transformers and its application in 3D -- NeRFs, and how INRs fit within the larger landscape of AI. While INRs inherently learn data distributions and do not require pretraining from other models, utilising INRs as a guide for other models presents an interesting direction, which we explore in more detail in Section 1 of the revised version.
>
> In addition, we want to remind the reviewer that this survey aims to explore the underlying technical principles that define Implicit Neural Representations (INRs), which form the foundation for many community-developed applications. By focusing on these core techniques, we provide a comprehensive comparison of methodologies, highlighting their strengths, limitations, while we acknowledge there are various applications based on these fundamental INRs have been developed within the community.
>
>
> ➡️ **Specific typo/error corrections for clarity**
>
> We thanks the reviewer for the detailed suggestions, the change has been made accordingly in blue in the revised verison.

---

> > ### Author Response · Authors · 2024-12-27
> > **Official Response 2**
> >
> > ➡️ **Broader Impact Concerns:
> > There is no included broader impact statement. One could be included (either separately or as a paragraph in the finally section) to strengthen the work, discussing topics on ethical issues surrounding INRs such as deepfakes.**
> >
> > Thank you for your suggestion regarding including a broader impact statement. We acknowledge the importance of discussing the potential ethical and societal implications of INR methodologies, especially in applications such as deepfakes and other sensitive areas.
> >
> > However, our work is intended as a survey of INR methodologies and applications rather than introducing new techniques, and broader impact statements are not typically included in surveys in this domain. This contrasts with the practice in some conferences, where such statements are a required element to address the implications of newly proposed methods. Nonetheless, we agree that discussing broader impacts could add value to the paper, and we added a couple of lines in the conclusion about this.

---

> > ### Comment · Reviewer_7Zht · 2024-12-30
> > **Thank you for response.**
> >
> > Thank you for your detailed response and revisions: most of them have strengthened this work.
> >
> > Regarding the new Section 2.5 and Table 1, this feels a bit odd to discuss complexity in Big-O notation when all of the activation functions are essentially the same. A much more useful analysis would be the real runtime (or other measure computational cost) of each algorithm for at least one of your experiments: for example, you could connect what you have added in Section 2.5 and Table 1 with the reported "Time" results in Table 3 with further discussion.
> >
> > Small grammatical errors in added revisions:
> > Page 1: "Recent works has shown..." -> "Recent works have shown..."
> > Page 11: " weight reprametisation" -> " weight reparametisation" (this one was a typo that I previously suggested to fix)

---

> > > ### Author Response · Authors · 2024-12-31
> > > **Official Response 3**
> > >
> > > ➡️ **Thank you for your detailed response and revisions: most of them have strengthened this work. Regarding the new Section 2.5 and Table 1, this feels a bit odd to discuss complexity in Big-O notation when all of the activation functions are essentially the same. A much more useful analysis would be the real runtime (or other measure computational cost) of each algorithm for at least one of your experiments: for example, you could connect what you have added in Section 2.5 and Table 1 with the reported "Time" results in Table 3 with further discussion.**
> > >
> > > We thank the reviewer for the insightful suggestion. We agree that connecting Section 2.5 and Table 1 with the runtime results reported in Table 3, accompanied by further discussion, would enhance the analysis and provide a more practical perspective on computational costs. We have revised the Section 2.5 in manuscript in blue.
> > >
> > > ➡️ **Small grammatical errors in added revisions: Page 1: "Recent works has shown..." -> "Recent works have shown..." Page 11: " weight reprametisation" -> " weight reparametisation" (this one was a typo that I previously suggested to fix)**
> > >
> > > We appreciate reviewer's careful attention to detail regarding grammatical errors in our revisions, we have updated the suggested modifications in blue.

---

### Review · Reviewer_Nozp · 2024-12-13

**Summary Of Contributions:**

This is a review paper on Implicit Neural Representations (INRs), which are neural networks used to model signals (e.g., images, 3D shapes and audio) as continuous functions. The paper starts by highlighting the advantages of IRNs, compared to existing methods. It then introduces a taxonomy that enables to compare the different INR methods based on their general structure. The taxonomies are: Activation function, positional encoding, activation function + positional encoding, and network structure.  Within these taxonomies, the paper summarizes key INR methods highlighting their advantages and disadvantages. In addition, the numerical experiment of the paper compares the different INR methods on different applied tasks.

**Audience:**

Yes

**Broader Impact Concerns:**

I have no concerns.

**Claims And Evidence:**

Yes

**Requested Changes:**

- The review recommends addressing some of the weaknesses listed above.

> Minor: Suggestions to improve writing/typesetting

- Page 2: The second sentence reads disjoint from the rest. Maybe combine the two sentences.

Specifically, Multi-Layer Perceptrons (MLPs) are trained to parameterise signals by taking input coordinates by a mapping technique that projects the network input into a higher-dimensional space. It predicts the associated data values.
- Full stop after equation 1
- Add full stop after "the final output reads: equation".
- Add comma after equations 3-9
- Be consistent: low-frequency bias or low frequency bias
- For equation (3), use math mode for sin
- Define i in (5) for the general reader
- Page 7 before equation 7:  which form is give by -->  which form is given by:
- Define Z, which appears in the set of translates, after equation 7
- In discussing Sinc function, be consistent with choice: Sinc or sinus cardinal
- Suggested rephrase before equation 8: To overcome this issue, the authors of the model
- Suggested rephrase before equation 9: Accordingly, the authors initialise the bias coefficients as:
- Define U in (9)
- After equation (11), exp should be typeset in math mode. Similar comment to cos and exp in (12) and (13)
- Also, add full stops after equations (12) and (13)
- Comma after equation (15)
- cos should be in math mode in equation 18
- Reviewer suggests rewriting the two sentences after equation (18) i.e., The sentence that starts with "Where"
- Add comma after equation (20)

**Strengths And Weaknesses:**

**Strengths**
- The paper is well-written, and easy to follow.
- The numerical section of the paper is strong, showing comparison of different INR methods across diverse tasks.
- I also appreciate the taxonomy proposed in the paper. Within this structure, it is easy to understand the different INR methods, for what data/signal they are suited for, and advantages/disadvantages.

**Weaknesses**
- I think the paper would benefit from defining more precisely computational complexity. In different places, computational complexity of different methods is referred as high or low. It is understandable that it might be hard to get exact computational complexity. I think the paper would benefit, at the very least, providing rough dependence of complexity on network and problem parameters. This will allow to at least appreciate why some methods are more expensive than others.
- There are few minor typesetting/writing issues that need to be corrected
- There are related methods which share similar themes. One example is latent representations (Autoencoders), where signals are encoded into low-dimensional latent spaces using neural networks (e.g., as autoencoders or variational autoencoders (VAEs)). Another set of methods is structured dictionary learning, where signals are approximated as sparse combinations of basis elements learned from data. Few references on the latter method are noted below. I think comparing these approaches from INR in the introduction section would be useful.

>References
- Chen, Yubei, Dylan Paiton, and Bruno Olshausen. "The sparse manifold transform." Advances in neural information processing systems 31 (2018).
- Tasissa, Abiy, et al. "K-Deep Simplex: Manifold Learning via Local Dictionaries." IEEE Transactions on Signal Processing (2023).
- Tschannen, Michael, Olivier Bachem, and Mario Lucic. "Recent Advances in Autoencoder-Based Representation Learning."

---

> ### Author Response · Authors · 2024-12-27
> **Official Response 1**
>
> **Weaknesses**
>
> ➡️ **I think the paper would benefit from defining more precisely computational complexity. In different places, computational complexity of different methods is referred as high or low. It is understandable that it might be hard to get exact computational complexity. I think the paper would benefit, at the very least, providing rough dependence of complexity on network and problem parameters. This will allow to at least appreciate why some methods are more expensive than others.**
>
> Thank you for your insightful suggestion. To address this comment, we have incorporated two  updates to the updatedpaper.
>
> Firstly, we have created a new table (now Table 1) that explicitly discusses the computational complexity of each INR method. This table includes the baseline complexity of $O(1)$ for all methods and details the additional computational operations that contribute to their overall cost, such as dynamic scaling, composition of functions, or division operations. This provides a clearer understanding of the relative computational requirements of different methods.
>
> Secondly, we have added a new subsection, now Section 2.5, that provides a detailed discussion of the baseline complexity alongside the additional operations for each method.  By including these dependencies, we aim to offer readers a rough estimate of the computational demands of each method and help them appreciate why some methods might be more computationally expensive than others.
>
> We believe these additions will clarify the relative computational costs of the INR methods.
>
>
> ➡️ **There are few minor typesetting/writing issues that need to be corrected**
>
> We thanks the reviewer for the detailed review, the addressed issue in Requested Changes has been updated in the revised manuscript.
>
> ➡️ **There are related methods which share similar themes. One example is latent representations (Autoencoders), where signals are encoded into low-dimensional latent spaces using neural networks (e.g., as autoencoders or variational autoencoders (VAEs)). Another set of methods is structured dictionary learning, where signals are approximated as sparse combinations of basis elements learned from data. Few references on the latter method are noted below. I think comparing these approaches from INR in the introduction section would be useful.**
>
> Thanks for suggestion the related methods, we add additional discussion of the technques that shown efficient representation of complex data in the Introduction in blue in the revised manuscript.
>
> **Requested Changes:**
>
> ➡️ **The review recommends addressing some of the weaknesses listed above. Minor: Suggestions to improve writing/typesetting.**
>
> We thanks the insightful suggestion that reviewer made in the weakness, and modified the manuscript with updates in blue.
>
> ➡️ **Page 2: The second sentence reads disjoint from the rest. Maybe combine the two sentences.
> And the following suggested changes.**
>
> We thanks the reviewer for the detailed suggestions, we have updated the manuscript with suggested modifications in blue.

---

> > ### Comment · Reviewer_Nozp · 2025-01-03
> > **Thank you for your response**
> >
> > Thank you for your response. I have checked the revised manuscript, and all my comments have been addressed.

---

### Review · Reviewer_64LX · 2024-12-18

**Summary Of Contributions:**

The paper presents a survey of the current state-of-the-art INR models. It develops a taxonomy for categorizing different INR approaches. These include four key areas: activation functions, positional encoding, combinations of these and network structure optimization. They analyze the trade-offs between current state-of-the-art INR techniques based on the properties of each as well as their own experiments.  The conclusions from the experiments are that INCODE and FR perform the best across nearly all INR tasks but require extensive computation. Finally, the paper highlights new avenues for improvement in all four of these key areas.

**Audience:**

Yes

**Broader Impact Concerns:**

None.

**Claims And Evidence:**

Yes

**Requested Changes:**

- Refine the statements on ReLU networks in light of [1].
- Include some discussion/experiments comparing INR methods to traditional image processing techniques.
- Writing issues brought up by the other reviewers.
- For the experiments, the authors should clarify whether they used the same default hyperparameters across all tasks or a different default for each task.
- One experimental setting where the hyperparameters are optimized for each INR method to provide a fair comparison.

**Strengths And Weaknesses:**

### Strengths
I think this is a good paper that attempts to unify a wide range of techniques for INRs. It brings a lot of new insight into how to compare different INR architectures and techniques and how to choose the appropriate methodology for a particular INR task. The paper does an excellent job of providing a taxonomy of the different methods and providing an empirical comparison of them.

### Weaknesses
- As a survey paper, I am curious about how INRs hold up against traditional image processing techniques. What are the advantages of an INR approach for CT reconstruction compared to standard TV regularized reconstruction?
- There has been some recent work showing that, despite conventional belief, ReLU neurons can be used in INRs without positional encodings [1]. That paper makes a clear distinction between optimizing the INR to fit an image and the expressive power of the INR. Since this work is claiming that it provides an "extensive review of INR methodologies and applications" it would be good to at least mention this work if not include it in the experiments.
- In light of the work mentioned above I feel the following statement, at the beginning of Section 2.1, should be refined.
    - "The ReLU function, while effective in many settings, has a discontinuous nature that limits its ability to capture higher-frequency signals, making it suboptimal for tasks that require detailed or high-resolution data representation."
    - The discontinuities are not the issue. Rather its the monotonic nature of the ReLU that creates an ill-conditioned loss landscape making it difficult to optimize the network via gradient descent.
- Regarding WIRE, the authors should also mention that another drawback is that it requires two hyper-parameters ($s$ and $\omega$) whereas other activation functions only require one. This in turn results in more computational effort to choose the appropriate hyperparameters.
- Many of these INR methodologies depend, crucially, on how the hyperparameters are chosen. However, while the experimental section uses the default hyperparameters from each method it is difficult to conclude whether some methods are truly better or just better tuned. In addition it is my understanding that there are different "default" hyperparameters depending on the task. For instance in WIRE the hyperparameters that work well for audio reconstruction are not necessarily expected to work for CT reconstruction or super-resolution. So were the same hyperparameters used across tasks or did the experiments use the default hyperparameters for each task?
- The paper does not consider or mention any work that tries to provide a theoretical understanding of why these methods work so well. For instance [2][3]. This might be worth mentioning these to make the survey more self-contained.
- (minor) There are a number of small writing issues that should be revised.

[1] Shenouda, Joseph, Yamin Zhou, and Robert D. Nowak. "ReLUs Are Sufficient for Learning Implicit Neural Representations." Forty-first International Conference on Machine Learning.

[2] Yüce, Gizem, et al. "A structured dictionary perspective on implicit neural representations." Proceedings of the IEEE/CVF Conference on Computer Vision and Pattern Recognition. 2022.

[3] Najaf, Mahrokh, and Gregory Ongie. "Towards a Sampling Theory for Implicit Neural Representations." arXiv preprint arXiv:2405.18410 (2024).

---

> ### Author Response · Authors · 2024-12-27
> **Official Response 1**
>
> **Weaknesses**
>
> ➡️ **As a survey paper, I am curious about how INRs hold up against traditional image processing techniques. What are the advantages of an INR approach for CT reconstruction compared to standard TV regularized reconstruction?**
>
> Thank you for your question. While our paper focuses on comparing implicit neural representation (INR) methodologies, we recognise the importance of understanding how INRs relate to traditional image processing techniques like total variation (TV)-regularised reconstruction.
>
> The primary advantage of INRs for CT reconstruction lies in their flexibility and adaptivity. Traditional TV-regularised reconstruction methods rely on explicit priors, such as sparsity in the gradient domain, which can effectively reduce noise and artifacts. However, these priors are handcrafted and may not fully capture the underlying complexity of the image, especially in cases where structures deviate from the assumed regularity or sparsity patterns.
>
> In contrast, INRs parameterise the image as a continuous function learned directly from the data. This enables them to model fine details and complex features without relying on fixed assumptions. For example, INR-based methods can adapt to varying frequency content in the data and inherently incorporate learned priors through their network architecture and optimisation. Additionally, INRs provide a natural framework for incorporating spatial continuity, which can be advantageous in scenarios like sparse-view CT reconstruction, where the data is undersampled or noisy.
>
> Another key difference is that INRs are resolution-agnostic. Once trained, an INR can generate reconstructions at arbitrary resolutions, whereas TV-based methods typically operate on fixed-resolution grids. This property makes INRs particularly suitable for applications like super-resolution or multi-scale imaging tasks, where fine details need to be reconstructed beyond the resolution of the input data.
>
> While INRs have demonstrated significant promise, it is worth noting that traditional methods like TV regularisation still hold advantages in terms of interpretability, computational efficiency, and robustness for certain tasks. Our survey focuses on INR methodologies and does not aim to benchmark INRs against traditional approaches-- as it will means another completely body of work due to the large options on explicit regularisation, but we hope this discussion provides some clarity on the conceptual and practical distinctions between these methodologies.
>
>
> ➡️ **There has been some recent work showing that, despite conventional belief, ReLU neurons can be used in INRs without positional encodings [1]. That paper makes a clear distinction between optimizing the INR to fit an image and the expressive power of the INR. Since this work is claiming that it provides an "extensive review of INR methodologies and applications" it would be good to at least mention this work if not include it in the experiments.
> [1] Shenouda, Joseph, Yamin Zhou, and Robert D. Nowak. "ReLUs Are Sufficient for Learning Implicit Neural Representations." Forty-first International Conference on Machine Learning.**
>
> Thank you for pointing out this relevant work. We acknowledge the significance of the study, "ReLUs Are Sufficient for Learning Implicit Neural Representations," as it directly contributes to the field of INRs. The authors address the spectral bias of ReLU neurons and propose a constrained ReLU-based network (BW-ReLU). We have updated the manuscript adding this reference.
>
>
>
> ➡️ **In light of the work mentioned above I feel the following statement, at the beginning of Section 2.1, should be refined."The ReLU function, while effective in many settings, has a discontinuous nature that limits its ability to capture higher-frequency signals, making it suboptimal for tasks that require detailed or high-resolution data representation."**
>
> We thank the reviewer for the suggestion. We have now rewording our statment to address this concern.

---

> > ### Author Response · Authors · 2024-12-27
> > **Official Response 2**
> >
> > ➡️ **The discontinuities are not the issue. Rather its the monotonic nature of the ReLU that creates an ill-conditioned loss landscape making it difficult to optimize the network via gradient descent.**
> >
> > Thank you for pointing out the impact of ReLU's monotonicity on creating an ill-conditioned loss landscape, which indeed hinders effective optimisation via gradient descent. This is an important factor that affects the ability of ReLU networks to learn high-frequency components effectively.
> >
> > We want to clarify that  the discontinuities in ReLU refers to its inability to smoothly represent functions with high-frequency components. Discontinuities in the derivative of ReLU (from 0 to 1) result in spectral bias, as noted in prior work, limiting the network's ability to learn fine-grained variations in data, especially in the absence of positional encodings or other modifications. This characteristic can be a fundamental bottleneck for tasks requiring smooth or detailed representations, alongside the optimisation challenges created by its monotonicity.
> >
> > That is, while the monotonic nature of ReLU indeed affects optimisation, the discontinuous nature of its derivative also plays a critical role in limiting its expressivity for tasks that require high-frequency detail. Together, these factors contribute to the challenges of using ReLU for implicit neural representations without additional modifications.
> >
> >
> > ➡️ **Regarding WIRE, the authors should also mention that another drawback is that it requires two hyper-parameters ($s$ and $\omega$) whereas other activation functions only require one. This in turn results in more computational effort to choose the appropriate hyperparameters.**
> >
> > Thanks for the suggestion. We have updated the manuscript accordigly.
> >
> >
> >
> > ➡️ **Many of these INR methodologies depend, crucially, on how the hyperparameters are chosen. However, while the experimental section uses the default hyperparameters from each method it is difficult to conclude whether some methods are truly better or just better tuned. In addition it is my understanding that there are different "default" hyperparameters depending on the task. For instance in WIRE the hyperparameters that work well for audio reconstruction are not necessarily expected to work for CT reconstruction or super-resolution. So were the same hyperparameters used across tasks or did the experiments use the default hyperparameters for each task?**
> >
> > Thank you for raising this important point. In our experiments, we used the default hyperparameters reported by the original authors for each task. This decision was made to ensure a fair comparison, as these hyperparameters represent the best parameters identified by the authors for their respective methods and tasks. By using the task-specific defaults, we aimed to evaluate each methodology under the conditions that maximise its performance, as established in the original works.
> >
> > We agree that the choice of hyperparameters can significantly impact performance, and this variability is an inherent challenge in comparing methodologies. However, using the best hyperparameters per task, as reported by the original authors, allows us to provide a more accurate assessment of each method’s capabilities in its intended application.
> >
> > We have added a clarifying note regarding this in the experimental section. The changes can be seen in blue colour.
> >
> >
> > ➡️ **The paper does not consider or mention any work that tries to provide a theoretical understanding of why these methods work so well. For instance [2][3]. This might be worth mentioning these to make the survey more self-contained.**
> >
> > Thank you for bringing these two works to our attention. We would like to clarify the context of those works.
> >
> > Regarding [3], "Towards a Sampling Theory for Implicit Neural Representations," the work provides a valuable theoretical understanding of INR methodologies, focusing on sampling and approximation theory. However, the study uses "Fourier Features" as a case study—a technique that we have already covered in our paper. While [3] deepens the theoretical understanding of Fourier-based methods, our scope is in existing INRs methods.
> >
> > As for [2], "A Structured Dictionary Perspective on Implicit Neural Representations," the paper does not introduce a new INR methodology. Instead, it draws an analogy between INRs and structured signal dictionaries, providing a novel perspective on how INRs can be interpreted within this framework. While this perspective is interesting, our survey focuses on methodologies and applications rather than theoretical analogie
> >
> >
> > ➡️ **(minor) There are a number of small writing issues that should be revised.**
> >
> > Thanks for the feedback. We have read through the paper.
> >
> >
> >
> > **Requested Changes:**
> >
> > Thank you for pointing out all the points, we have updated the manuscript as addressed in the Weaknesses.

---

### Decision · Action_Editor_dru6 · 2025-02-04

**Recommendation:** Accept as is

**Comment:**

Based on the detailed reviews and discussion, I recommend accepting this paper for publication in TMLR with Survey Certification. The paper presents a comprehensive survey of Implicit Neural Representations (INRs) that is thoroughly supported by evidence. All three expert reviewers recommended acceptance and praised its value for both novice and experienced researchers in the field. The authors were responsive to reviewer requests, strengthening the paper in their revision.

The paper's strong technical quality and broad appeal to TMLR's audience are evident in its comprehensive taxonomy, thorough experimental comparisons, and clear practical guidance. The reviewers highlighted different strengths: unification of INR techniques, strong numerical experiments, and thorough literature coverage respectively.

**Audience:**

Yes, there is strong evidence that this paper would be of significant interest to TMLR's audience. The reviewers specifically noted the paper provides valuable insights for both novice and experienced researchers in the field. The focus on core machine learning topics like neural architectures, representations and optimization aligns with TMLR's scope, while the technical depth and implementation details match TMLR readers' interests.

**Claims And Evidence:**

The paper's claims are well-supported by accurate, convincing and clear evidence. The authors provide extensive experimental comparisons across multiple tasks. The paper's technical accuracy was verified by multiple expert reviewers. When asked to strengthen the evidence, the authors were very responsive in their revision. All three reviewers confirmed the sufficiency of evidence and technical soundness.